# GR-LoRA: Gradient-Recycling Low-Rank Adaptation for Class-Incremental Learning

Yipeng Lin [* 1]  Fengqiang Wan [* 1]  Yang Yang [1]

## Abstract

Pre-trained models with parameter-efficient fine-tuning have shown strong effectiveness in Class-Incremental Learning (CIL), which seeks to balance model plasticity and stability. In this context, orthogonality constraints can significantly enhance model stability, yet their reliance on subspace inevitably compromises model plasticity over long tasks. To address this, we propose Gradient-Recycling Low-Rank Adaptation (GR-LoRA), which reconciles stability and plasticity by recycling the gradients discarded in orthogonal projection. Specifically, GR-LoRA recycles post-decomposition non-orthogonal gradient components into task-specific lightweight modules and selects optimal module via entropy to improve plasticity, while incorporating local and global mismatch suppression to preserve stability by synthesizing out-of-distribution representations across all tasks. Theoretical analysis confirms that this recycling strategy preserves stability and improves plasticity. Experimental results from multiple CIL benchmarks verify the effectiveness and general applicability of GR-LoRA.

## 1. Introduction

Class-Incremental Learning (CIL) aims to enable models to progressively acquire knowledge from sequential data streams in real-world scenarios (De Lange et al., 2021; Parisi et al., 2019; Zhou et al., 2022). The primary challenge in CIL is stability-plasticity dilemma (Mermillod et al., 2013; CHEN et al., 2023), where stability characterizes the preservation of prior knowledge, and plasticity reflects the capacity to adapt to new concepts. Large-scale pre-trained models (PTMs) (Steiner et al., 2022; Radford et al., 2021),

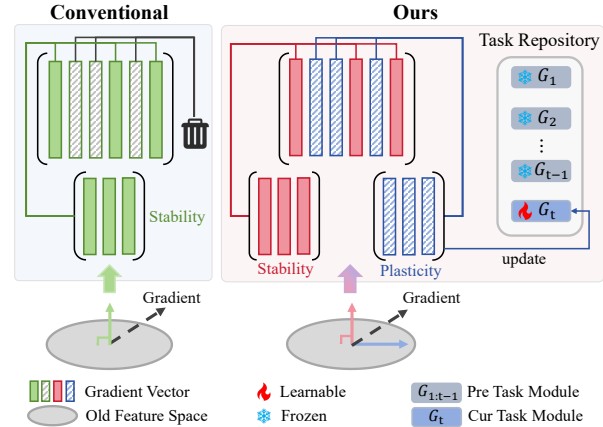

Figure 1. Left: Projection-based method enforces orthogonality to past task subspaces, discarding non-orthogonal gradient components. Right: Our method preserves the orthogonal update in the shared model, while routing the non-orthogonal component into a task repository implemented as task-specific modules.

together with parameter-efficient fine-tuning (PEFT) methods (Houlsby et al., 2019; Han et al., 2024) facilitate this dilemma by leveraging strong representations while updating only a minimal set of model parameters.

However, existing PTM-based CIL methods still grapple with distinct limitations. In the initial exploration, selection-based methods (Wang et al., 2022b; Smith et al., 2023; Wang et al., 2022a) dynamically select task-specific modules from a shared pool using input similarity, which can lead to erroneous selection due to overconfident matching (Pan, 2019). To circumvent unreliable selection, prototype-rectification methods (Zhou et al., 2024; Wu et al., 2025) adopt an expandable parameter learning strategy on the same model, update historical prototypes using current-task samples and rely on the rectified prototypes at inference time, which inevitably accumulates errors over long tasks. To avoid accumulated errors of prototype rectification, orthogonality-based methods (Liang & Li, 2024; Liu & Chang, 2025) project current-task gradients onto subspaces orthogonal to past tasks to prevent interference, but explicitly discarding the non-orthogonal components, as illustrated in Figure 1 (left). Consequently, as tasks accumulate, this constraint progressively contracts the feasible optimization space (Liang

*Equal contribution [1]Nanjing University of Science and Technology, Nanjing, China. Correspondence to: Yang Yang <yyang@njust.edu.cn>.

*Proceedings of the 43rd International Conference on Machine Learning*, Seoul, South Korea. PMLR 306, 2026. Copyright 2026 by the author(s).

& Li, 2024). As established in Theorem 3.1, this contraction results in reduced plasticity due to the elimination of non-orthogonal gradient components during within-task optimization. This observation motivates an investigation into whether the non-orthogonal gradient components can be explicitly repurposed to balance stability and plasticity.

Motivated by this, we propose Gradient Recycling Low-Rank Adaptation (GR-LoRA), which redirects the non-orthogonal gradient components into task-specific module, as illustrated in Figure 1(right). Specifically, adhering to orthogonality principles, GR-LoRA decomposes task gradients into orthogonal and non-orthogonal components, encodes the latter into task-specific LoRA (Hu et al., 2022) modules and selects optimal module via entropy to improve plasticity without compromising stability. Theorem 3.2 demonstrates that preserving these components in auxiliary parameter suffices to recover the original optimization trajectory. To further improve the reliability of this selection for preserving stability, we incorporate two suppression strategies, consisting of Local Mismatch Suppression (LMS) and Global Mismatch Suppression (GMS). LMS suppresses spurious activations by routing current-task samples through mismatched LoRA modules, forcing alignment with current categories. Subsequently, GMS leverages prototypes to synthesize out-of-distribution representations via inter-class similarity compensation. These synthetic features approximate the global distribution, enabling the classification head to effectively suppress mismatched signals across the entire tasks. Extensive experiments across diverse benchmark datasets demonstrate that GR-LoRA outperforms existing state-of-the-art (SOTA) approaches.

## 2. Related Work

### 2.1. PTM-Based Class-Incremental Learning

The advent of large-scale PTM (Steiner et al., 2022; Radford et al., 2021) has provided robust and generalizable representations for downstream tasks. However, fully fine-tuning these models is computationally prohibitive and prone to catastrophic forgetting (French, 1999; French & Ferrara, 2020). Consequently, PEFT (Han et al., 2024; Houlsby et al., 2019) has established itself as the mainstream approach for CIL. By freezing the pre-trained backbone and optimizing only a minimal set of parameters, PEFT effectively preserves historical knowledge while accommodating new tasks. Based on the structural integration of trainable parameters, existing approaches generally fall into three categories: prompt-based, adapter-based, and LoRA-based methods. Prompt-based methods insert learnable tokens to encode knowledge via key-query retrieval mechanisms, such as L2P (Wang et al., 2022b) selects dynamic prompts from a pool to instruct the frozen backbone; DualPrompt (Wang et al., 2022a) utilizes general and expert prompts to separate task-invariant knowledge from task-specific ones; CODA-Prompt (Smith et al., 2023) employs a decomposed attention scheme to assemble prompts as weighted linear combinations of pool components. Adapter-based methods, such as EASE (Zhou et al., 2024) mitigates feature degradation by integrating multiple adapter predictions with semantic-guided prototype synthesis; SSIAT (Tan et al., 2024) aligns new and old task features by continuously tuning shared adapters and estimating mean shifts; and MOS (Sun et al., 2025) optimizes inference efficiency by merging adapter parameters via a self-optimization retrieval mechanism. LoRA-based methods, such as InfLoRA (Liang & Li, 2024) minimizes inter-task interference by imposing orthogonal constraints to isolate low-rank subspaces; CL-LoRA (He et al., 2025) splits the adaptation process by dedicating initial blocks to shared feature learning via early-exit distillation, while employing gradient reassignment on deeper blocks to preserve task-specific knowledge; LoRA-DRS (Liu & Chang, 2025) maintains representation stability by explicitly subtracting task-specific variations to construct a drift-resistant space; and MACIL (Wu et al., 2025) mitigates semantic drift by incorporating mean shift compensation and covariance calibration to align class representations across tasks. Despite these advances, existing PTM-based CIL methods still struggle with stability-plasticity dilemma. We propose GR-LoRA, a novel orthogonality-based low-rank adaptation method that simultaneously ensuring stability and enhancing plasticity through adaptive recycle gradient.

### 2.2. Out-of-Distribution Detection

Out-of-Distribution (OOD) detection (Sun et al., 2022; Yang & Xu, 2025) is critical for ensuring model reliability by identifying samples deviating from the training distribution . Existing approaches generally fall into two paradigms: post-hoc scoring and training-time regularization. Post-hoc methods derive uncertainty scores from pre-trained classifiers. Baseline metrics, such as entropy (Liu et al., 2020), operate on the assumption that models exhibit higher confidence for In-Distribution (ID) samples than OOD samples. In modular CIL, where distinct task-specific models are maintained, identifying the correct module is intrinsically an OOD problem (Aljundi et al., 2017). Specifically, TUNA (Wang et al., 2025) has demonstrated that entropy minimization can effectively select the appropriate task-specific expert during inference. Conversely, training-time methods modify the optimization objective to enforce compact decision boundaries. A seminal approach is Outlier Exposure (Hendrycks et al., 2019; Du et al., 2022), which leverages large-scale auxiliary datasets to explicitly penalize the model's confidence on anomalies. Adapting this to CIL, methods such as TPL (Lin et al., 2024) leverage stored data from past tasks as surrogate OOD classes, thereby reinforcing the boundary discrimination between current and historical task.

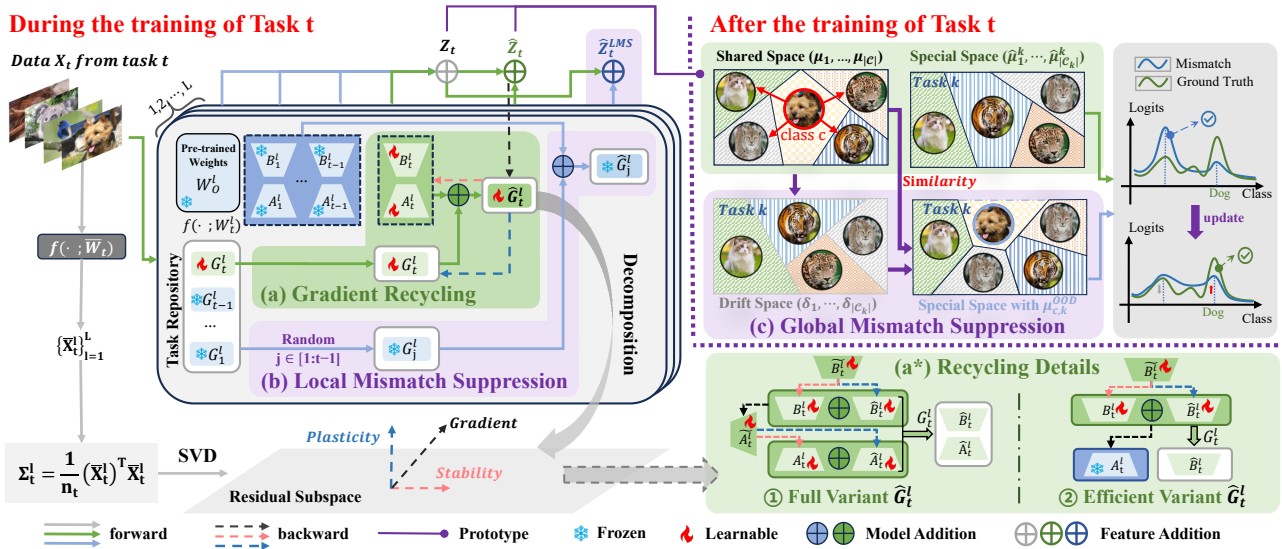

**Figure 2.** Illustration of the proposed framework. Before training on task $t$, a residual subspace is constructed from the current task data. During training of task $t$, the pre-trained backbone and all historical LoRA modules are frozen. We apply **(a) Gradient Recycling**: the learnable modules $G_t$, $A_t$, and $B_t$ are aggregated to form $\hat{G}_t$ in the forward pass, while the backward gradients are projected onto the residual subspace and decomposed, as detailed in **(a*) Recycling Details**. Simultaneously, **(b) Local Mismatch Suppression** is employed by feeding the current data through historical modules. After training on task $t$, **(c) Global Mismatch Suppression** leverages statistical class prototypes to synthesize out-of-distribution features together with in-distribution features for classifier retraining.

## 3. Method

### 3.1. Preliminaries

The CIL problem is formulated as a stream of $T$ tasks with disjoint label spaces. Let $\mathcal{D}_t = \{(\mathbf{x}_{t,i}, y_{t,i})\}_{i=1}^{N_t}$, denote the dataset for the $t$-th task, where $\mathbf{x}_{t,i} \in \mathcal{X}_t$ and $y_{t,i} \in \mathcal{Y}_t$, and the label spaces satisfy $\mathcal{Y}_t \cap \mathcal{Y}_{t'} = \emptyset$ for all $t \neq t'$. Access to historical data $\mathcal{D}_{1:t-1}$ is strictly prohibited.

**During the training of task t.** For each task $t$, the pre-trained backbone $W_0$ and all accumulated task parameters $\{B_j A_j\}_{j=1}^{t-1}$ are frozen. A new trainable low-rank branch $\Delta W_t = B_t A_t$ is introduced, with $A_t \in \mathbb{R}^{r \times d}$ initialized to zero and $B_t \in \mathbb{R}^{d \times r}$ initialized from a Gaussian distribution, with rank $r \ll d$. Consequently, the weight matrix at layer $l$ for the task $t$ is given by

$$W_t^l = W_{t-1}^l + B_t^l A_t^l = W_0^l + \sum_{j=1}^{t} B_j^l A_j^l. \quad (1)$$

Given the input $X_t = \{x_{t,i}\}_{i=1}^B \in \mathcal{X}_t$, the network output $Z_t$ is obtained via layer-wise propagation as $Z_t^l = f(X_t^l; W_t^l)$, $X_t^{l+1} = \sigma(Z_t^l)$, where $\sigma(\cdot)$ is nonlinear activation, and $l \in \{1, \ldots, L\}$, $X_t^1 = X_t$. The model is optimized by minimizing the Cross Entropy (CE) loss:

$$\mathcal{L}'_{CE} = CE(Z_t, Y_t), \quad (2)$$

where $Y_t \in \mathcal{Y}_t$ denotes the corresponding labels.

**After the training of task t.** Following (Tan et al., 2024; Wu et al., 2025), the unified classifier is retrained after

each task using pseudo-features sampled from Gaussian distributions of all learned classes. Specifically, at each task, we compute class prototypes $\mu_c$ and covariance matrices $\Sigma_c$ for all classes $c \in \mathcal{Y}_t$ based on the current feature $Z_t$. We then draw $s_c$ pseudo-features $h^c \sim \mathcal{N}(\mu_c, \Sigma_c)$ for each class. The classification head is subsequently optimized using the CE loss on these synthetic samples:

$$\mathcal{L}_{CA} = -\frac{1}{s_c|\mathcal{C}|} \sum_{c=1}^{|\mathcal{C}|} \sum_{i=1}^{s_c} CE(h_i^c, c) \quad (3)$$

where $\mathcal{C}$ denotes the set of all accumulated classes.

### 3.2. Overview

As illustrated in Figure 2, our framework follows a two-stage optimization scheme (Zhang et al., 2023a). During the training of task $t$, we employ Gradient Recycling alongside Local Mismatch Suppression (LMS). However, the accumulation of non-orthogonal gradients within task-specific modules inevitably biases the classifier. To rectify this, after the training of task t, we employ Classifier Alignment (CA) using generated samples. Within this phase, we incorporate Global Mismatch Suppression (GMS), which synthesizes OOD prototypes from ID statistics to assist CA in recalibrating global decision boundaries.

### 3.3. Gradient Recycling

In this section, we introduce Gradient Recycling, which recycles the gradients discarded by orthogonal constraints

(Liang & Li, 2024; Liu & Chang, 2025) into a task-specific lightweight modules, enhancing plasticity of current task.

**Residual Subspace Construction.** Before learning task $t$, the feature space of the new task is isolated by temporarily removing the accumulated LoRA parameters from previous tasks (Liu & Chang, 2025). Specifically, the effective backbone weights at layer $l$ are adjusted as $\bar{\mathbf{W}}_t^l = \mathbf{W}_0^l - \sum_{j=1}^{t-1} \mathbf{B}_j^l \mathbf{A}_j^l$. Forwarding the current task data $\mathcal{D}_t$ through this subtraction-adjusted model yields layer-wise features $\bar{\mathbf{X}}_t^l$. The empirical covariance matrix is then computed as $\mathbf{\Sigma}_t^l = \frac{1}{N_t}(\bar{\mathbf{X}}_t^l)^\top \bar{\mathbf{X}}_t^l$, where $N_t$ denotes the number of samples in task $t$. Principal directions of the residual feature space are obtained via singular value decomposition (SVD) (Baker, 2005), $\mathbf{U}_t^l \mathbf{\Lambda}_t^l (\mathbf{U}_t^l)^\top = \mathrm{SVD}(\mathbf{\Sigma}_t^l)$, where $\mathbf{U}_t^l$ contains orthogonal eigenvectors and $\mathbf{\Lambda}_t^l$ represents the associated singular values. Based on these principal directions, residual subspace basis at layer $l$ is defined by the top-$k$ components, $\mathbf{P}_t^l = (\mathbf{U}_t^l)_k$. Accordingly, the matrix $\mathbf{P}_t^l(\mathbf{P}_t^l)^\top$ defines an orthogonal projector onto the residual subspace, which restricts parameter updates to directions that minimally interfere with previously learned tasks.

**Gradient Recycling Optimization.** Based on projector $\mathbf{P}_t^l(\mathbf{P}_t^l)^\top$, optimization for task $t$ is performed through Gradient Recycling Module $\hat{G}_t$, as shown in Figure 2(a). Let $\mathbf{g}_{t,s}^l$ denote gradient at layer $l$ and training step $s$, and gradient decomposition induced by projection is given by

$$\mathbf{g}_{t,s}^l = \mathbf{P}_t^l(\mathbf{P}_t^l)^\top \mathbf{g}_{t,s}^l + \left(\mathbf{I} - \mathbf{P}_t^l(\mathbf{P}_t^l)^\top\right) \mathbf{g}_{t,s}^l, \quad (4)$$

where $\mathbf{I}$ denotes the identity matrix, and the two terms correspond to orthogonal components responsible for stability and residual components accounting for plasticity.

$\hat{G}_t$ is implemented via a dual-component LoRA architecture, as shown in Figure 2(a*) full variant. The effective LoRA parameters are decomposed into a shared orthogonal component $\{\mathbf{A}_t^l, \mathbf{B}_t^l\}$ and a task-specific component $\{\hat{\mathbf{A}}_t^l, \hat{\mathbf{B}}_t^l\}$. The forward mapping is given by

$$\begin{aligned} \hat{W}_t^l &= W_0^l + \sum_{j=1}^{t-1} \mathbf{B}_j^l \mathbf{A}_j^l + (\mathbf{B}_t^l + \hat{\mathbf{B}}_t^l)(\mathbf{A}_t^l + \hat{\mathbf{A}}_t^l) \\ &= W_{t-1}^l + \tilde{\mathbf{B}}_t^l \tilde{\mathbf{A}}_t^l = W_{t-1}^l + \hat{G}_t^l, \end{aligned} \quad (5)$$

where $\tilde{\mathbf{B}}_t^l$ and $\tilde{\mathbf{A}}_t^l$ denote the effective unconstrained LoRA parameters. The task-specific components $\hat{\mathbf{A}}_t^l$ and $\hat{\mathbf{B}}_t^l$ are initialized to zero, ensuring that the initial training dynamics coincide with standard LoRA (Hu et al., 2022). By gradient additivity (Rahel & Hubert, 1991; Müller & Yao, 2010), the orthogonal component $\mathbf{P}_t^l(\mathbf{P}_t^l)^\top \mathbf{g}_{t,s}^l$ is assigned to the shared parameter module $\{\mathbf{A}_t^l, \mathbf{B}_t^l\}$ to preserve stability across tasks, while the residual component $(\mathbf{I} - \mathbf{P}_t^l(\mathbf{P}_t^l)^\top)\mathbf{g}_{t,s}^l$ is assigned to the task-specific module $\{\hat{\mathbf{A}}_t^l, \hat{\mathbf{B}}_t^l\}$ to facilitate plasticity for the current task. Sum-

ming both updates recovers the original unconstrained gradient, with interference confined to the task-specific module.

To improve training efficiency, a simplified variant of $\hat{G}_t$ is adopted, as illustrated in Figure 2(a*). Exploiting the commutativity of low-rank updates under mild conditions (Zhang et al., 2023b), the input projection $A_t$ is frozen and only the task-specific output projection $\hat{B}_t$ is learned, which is mathematically equivalent to the full formulation while substantially reducing computational cost. The LoRA weight formulation simplifies to:

$$\begin{aligned} \hat{W}_t^l &= W_0^l + \sum_{j=1}^{t-1} B_j^l A_j^l + (B_t^l + \hat{B}_t^l) \cdot A_t^l \\ &= W_{t-1}^l + \tilde{B}_t^l A_t^l = W_{t-1}^l + \hat{G}_t^l. \end{aligned} \quad (6)$$

Regardless of the architectural variant, model optimization is performed using the CE loss:

$$\mathcal{L}_{CE} = CE(\hat{Z}_t, Y^t), \quad (7)$$

where $\hat{Z}_t = f(X^t; \hat{W}_t)$ denotes the model output, and $\hat{W}_t$ denotes the collection of parameters across all layers. After training, task-specific components are consolidated into the task repository, represented as $G_t = \{\hat{B}_t, \hat{A}_t\}$ for the full variant and $G_t = \{\hat{B}_t\}$ for the efficient variant.

### 3.4. Mismatch Suppression

Upon the completion of task, the system must identify the optimal task-specific LoRA module for each input $x$ without access to task identity (Wan & Yang, 2025). Let $\tilde{G}_j$ denote the composite weight matrix for the $j$-th task, defined as $\tilde{G}_j = W_0^l + \sum_{j'=1, j'\neq j}^{t} B_{j'}^l A_{j'}^l + \hat{G}_j^l$. A critical challenge in CIL is that historical modules often exhibit high confidence on current task data due to semantic overlap, acting as an OOD disturbance (Xu & Yang, 2025). To mitigate this, we employ entropy as a selection metric (Wang et al., 2025). The optimal module is determined by minimizing the prediction entropy:

$$\tilde{G}^* = \underset{\tilde{G}_j \in \{\tilde{G}_1, \dots, \tilde{G}_t\}}{\arg\min} - \sum_{c=1}^{\mathcal{C}} f_c(x; \tilde{G}_j) \log f_c(x; \tilde{G}_j), \quad (8)$$

where $f_c(x; \tilde{G}_j)$ denotes the predicted probability of class $c$ given input $x$ and $\mathcal{C}$ denotes all accumulated classes.

However, sole reliance on intrinsic entropy is insufficient, as mismatched historical modules may still exhibit high confidence on current task data. To address this, we propose two Mismatch Suppression strategies, comprising LMS and GMS. As illustrated in Figure 2, LMS is integrated into the training phase of task $t$ while GMS is applied during the classifier alignment phase after task $t$.

**Local Mismatch Suppression.** To enhance the discriminability of the selection metric during the training

phase, we propose LMS, which integrates an Outlier Exposure (Hendrycks et al., 2019; Miao et al., 2024) mechanism in training phase. As illustrated in Figure 2(b), LMS treats current data $X^t$ as OOD samples relative to all historical modules $\{\tilde{G}_j\}_{j=1}^{t-1}$. Crucially, instead of enforcing a generic uniform distribution that risks degrading ID feature discriminability, we utilize the ground truth label as a directional OOD target. This allows us to employ the standard CE loss to concentrate probability mass onto this foreign class, effectively suppressing activations on the module's native classes while preserving historical performance. To reduce the computational overhead of traversing all historical modules, we uniformly sample a single mismatched module index $j$ per mini-batch to impose this constraint. Formally, given the prediction $\hat{Z}_t^{\text{LMS}} = f(X^t; \tilde{G}_j)$, the LMS objective for task $t$ is given by: $\mathcal{L}_{LMS} = CE(\hat{Z}_t^{\text{LMS}}, Y_t)$. Consequently, the total loss for during the training of task $t$ is formulated as:

$$L_{total} = L_{CE} + L_{LMS}. \tag{9}$$

**Global Mismatch Suppression.** While LMS effectively prevents historical modules from overfitting to the current data, it provides only local regularization. It fails to capture the converse failure mode, where task-specific modules of the current task over-generalize to classes from previous tasks, leading to global task confusion. To address this limitation, we propose GMS, which synthesizes virtual OOD features across all task-specific modules to enforce a globally consistent decision boundary, as shown in Figure 2(c). Specifically, let $\mu_c$ denote the prototype of class $c$ in the shared space, while $\hat{\mu}_c$ and $\hat{\Sigma}_c$ denote its prototype and covariance in the task-specific subspace, respectively. The prototype drift vector as $\delta_c = \hat{\mu}_c - \mu_c$, which captures the geometric shift between the two spaces (Zhou et al., 2024; Fukuda et al., 2025; Li et al., 2025). Based on these statistics and inspiration, we first compute the semantic affinity $\alpha_{c',c}$ between the target class $c$ and the task's native classes $c' \in \mathcal{C}_k$ using cosine similarity in the shared space:

$$\alpha_{c',c} = \frac{\exp(\text{sim}(\mu_{c'}, \mu_c)/\tau)}{\sum_{z \in \mathcal{C}_k} \exp(\text{sim}(\mu_z, \mu_c)/\tau)}. \tag{10}$$

Next, we estimate the virtual OOD prototype $\mu_{c,k}^{OOD}$ by transferring the weighted geometric drift of these native classes to the target class $c$:

$$\mu_{c,k}^{OOD} = \mu_c + \sum_{c' \in \mathcal{C}_k} \alpha_{c',c} \cdot \delta_{c'}. \tag{11}$$

This process is extended globally to construct a set of OOD prototypes $P_c^{OOD} = \{\mu_{c,k}^{OOD} \mid k \in [1,t], k \neq Task(c)\}$ for every class $c$ across all disjoint task subspaces.

Finally, to align the classifier, we sample $s_c/t$ pseudo-OOD samples $h^{c,k} \sim \mathcal{N}(\mu_{c,k}^{OOD}, \hat{\Sigma}_c)$ for each task $k$ subspace.

The classification head is retrained using the CE loss to enforce consistent prediction across different task subspaces:

$$\mathcal{L}_{GMS} = -\frac{1}{s_c|\mathcal{C}|} \sum_{c=1}^{|\mathcal{C}|} \sum_{i=1}^{s_c/t} \sum_{k=1}^{t} CE(h_i^{c,k}, c). \tag{12}$$

By further incorporating the standard classifier alignment loss on available data, the overall objective for classifier retraining is given by:

$$\mathcal{L}_{head} = \mathcal{L}_{CA} + \mathcal{L}_{GMS}. \tag{13}$$

### 3.5. Theoretical Analysis

To analyze the plasticity of the proposed method, we study the attainable population risk under different parameter constraints from an optimization perspective. We assume that the population risk $R_t(w)$ is $\mu$-strongly convex and $L$-smooth with respect to the model parameters.

**Theorem 3.1.** *For each task $t = 1, \dots, T$, let $\widehat{\mathbf{W}}_{\perp,t}$ denote the solution obtained under the orthogonal constraint induced by the layer-wise projectors $\{\Pi_t^l\}_{l=1}^L$, and let $\mathbf{W}_t^\star$ be the unconstrained population minimizer of $R_t$. Then, with probability at least $1 - \delta$, it holds that*

$$\sum_{t=1}^{T} \left( R_t(\widehat{\mathbf{W}}_{\perp,t}) - R_t(\mathbf{W}_t^\star) \right) \leq \sum_{t=1}^{T} \sum_{l=1}^{L} \left( 4 \Re_{n_t}(\mathcal{F}_{\perp,t}^l) \right.$$
$$\left. + \frac{1}{2\mu} \|\mathbf{Q}_t^l \nabla_{\mathbf{W}^l} R_t(\mathbf{W}_{\perp,t}^\star)\|_F^2 + 2\sqrt{\frac{\log(2TL/\delta)}{2n_t}} \right), \tag{14}$$

*where $\mathbf{W}_{\perp,t}^\star$ is the population minimizer under orthogonal constraint and $\Re_{n_t}(\cdot)$ denotes the Rademacher complexity.*

As established in Theorem 3.1, when model updates are constrained to orthogonal subspace, the cumulative performance gap relative to the unconstrained optimum is dominated by $\|\mathbf{Q}_t^l \nabla_{\mathbf{W}^l} R_t(\mathbf{W}_{\perp,t}^\star)\|_F^2$. Consequently, even under perfect samples and optimization, orthogonal constraints induce an unavoidable plasticity gap.

**Theorem 3.2.** *Suppose the model parameters are augmented in the LoRA form $\mathbf{W}^l = \mathbf{W}_0^l + \sum_{j=1}^{t-1} \mathbf{B}_j^l \mathbf{A}_j^l + (\mathbf{B}_t^l + \hat{\mathbf{B}}_t^l)(\mathbf{A}_t^l + \hat{\mathbf{A}}_t^l)$, where $\mathbf{B}_t^l \mathbf{A}_t^l$ denotes the shared module learned under orthogonal constraints, and $\hat{\mathbf{B}}_t^l \hat{\mathbf{A}}_t^l$ is a task-specific module allocated to task $t$. Let $\mathbf{W}_{\text{GR},t}^\star$ denote the population minimizer under this parameterization. Then the cumulative optimality gap across tasks satisfies*

$$\sum_{t=1}^{T} \left( R_t(\mathbf{W}_{\text{GR},t}^\star) - R_t(\mathbf{W}_t^\star) \right) \leq \sum_{t=1}^{T} \sum_{l=1}^{L}$$
$$\frac{1}{2\mu} \|\mathbf{Q}_t^l \nabla_{\mathbf{W}^l} R_t(\mathbf{W}_{\perp,t}^\star) - \Pi_{\mathcal{S}_t^l}(\mathbf{Q}_t^l \nabla_{\mathbf{W}^l} R_t(\mathbf{W}_{\perp,t}^\star))\|_F^2, \tag{15}$$

*where $\Pi_{\mathcal{S}_t^l}$ denotes the projection onto $\mathcal{S}_t^l = \text{range}(\hat{\mathbf{B}}_t^l \hat{\mathbf{A}}_t^l)$.*

*Table 1.* Last and average performance results on CIFAR-100 and ImageNet-R under the long-term setting (20 and 50 tasks). The mean and standard deviation of three trials are provided. We compare all methods using the same ViT-B/16-IN21K backbone, seeds, and class orders. Red denotes the best result and blue denotes the second-best result in each column.

| Method | CIFAR-100 | | | | ImageNet-R | | | |
|---|---|---|---|---|---|---|---|---|
| | 20 | | 50 | | 20 | | 50 | |
| | $\mathcal{A}_{Last}$ | $\mathcal{A}_{Avg}$ | $\mathcal{A}_{Last}$ | $\mathcal{A}_{Avg}$ | $\mathcal{A}_{Last}$ | $\mathcal{A}_{Avg}$ | $\mathcal{A}_{Last}$ | $\mathcal{A}_{Avg}$ |
| L2P (Wang et al., 2022b) | $79.51_{\pm0.67}$ | $85.50_{\pm1.23}$ | $73.91_{\pm1.67}$ | $81.90_{\pm0.98}$ | $69.64_{\pm0.42}$ | $75.28_{\pm0.57}$ | $55.89_{\pm1.59}$ | $62.98_{\pm2.89}$ |
| DualPrompt (Wang et al., 2022a) | $80.44_{\pm1.38}$ | $86.96_{\pm1.98}$ | $76.66_{\pm0.74}$ | $85.18_{\pm0.92}$ | $66.61_{\pm0.58}$ | $72.45_{\pm0.37}$ | $61.50_{\pm0.86}$ | $68.63_{\pm1.31}$ |
| CODA-Prompt (Smith et al., 2023) | $81.36_{\pm0.88}$ | $88.17_{\pm0.61}$ | $55.45_{\pm0.48}$ | $68.39_{\pm0.53}$ | $69.96_{\pm0.50}$ | $75.34_{\pm0.85}$ | $48.89_{\pm0.90}$ | $55.59_{\pm2.67}$ |
| SLCA (Zhang et al., 2023a) | $89.62_{\pm0.18}$ | $93.03_{\pm1.09}$ | $87.90_{\pm0.17}$ | $91.96_{\pm1.30}$ | $75.53_{\pm0.42}$ | $80.65_{\pm1.16}$ | $68.95_{\pm4.46}$ | $71.15_{\pm10.57}$ |
| InfLoRA (Liang & Li, 2024) | $81.59_{\pm0.94}$ | $87.33_{\pm2.32}$ | $55.19_{\pm2.83}$ | $69.96_{\pm3.55}$ | $73.01_{\pm0.82}$ | $79.76_{\pm0.83}$ | $61.91_{\pm1.31}$ | $71.20_{\pm0.54}$ |
| EASE (Zhou et al., 2024) | $86.32_{\pm0.48}$ | $90.83_{\pm1.14}$ | $77.43_{\pm3.11}$ | $84.59_{\pm1.22}$ | $73.78_{\pm0.47}$ | $80.28_{\pm0.67}$ | $68.53_{\pm0.04}$ | $75.49_{\pm1.06}$ |
| SSIAT (Tan et al., 2024) | $90.07_{\pm0.56}$ | $93.54_{\pm0.83}$ | $87.33_{\pm1.46}$ | $91.63_{\pm0.89}$ | $78.31_{\pm0.53}$ | $82.35_{\pm0.52}$ | $74.52_{\pm0.35}$ | $79.01_{\pm0.54}$ |
| CL-LoRA (He et al., 2025) | $83.75_{\pm1.39}$ | $88.98_{\pm1.87}$ | $71.14_{\pm2.44}$ | $78.80_{\pm2.72}$ | $77.20_{\pm0.66}$ | $83.45_{\pm0.56}$ | $68.36_{\pm0.82}$ | $76.80_{\pm1.27}$ |
| LoRA-DRS (Liu & Chang, 2025) | $88.76_{\pm0.22}$ | $92.35_{\pm0.90}$ | $86.51_{\pm1.05}$ | $91.17_{\pm0.97}$ | $74.96_{\pm0.17}$ | $80.66_{\pm0.67}$ | $72.17_{\pm0.27}$ | $78.05_{\pm0.70}$ |
| MOS (Sun et al., 2025) | $89.48_{\pm0.39}$ | $92.97_{\pm1.01}$ | $87.05_{\pm0.90}$ | $91.44_{\pm1.21}$ | $75.04_{\pm0.75}$ | $80.39_{\pm0.87}$ | $66.92_{\pm0.34}$ | $74.81_{\pm0.41}$ |
| MACIL (Wu et al., 2025) | $90.31_{\pm0.17}$ | $93.47_{\pm0.78}$ | $85.09_{\pm0.87}$ | $90.89_{\pm1.24}$ | $79.46_{\pm0.15}$ | $84.25_{\pm0.51}$ | $70.10_{\pm1.02}$ | $77.47_{\pm0.55}$ |
| **E-GR-LoRA** | $91.14_{\pm0.02}$ | $94.11_{\pm0.86}$ | $89.76_{\pm0.23}$ | $93.16_{\pm0.77}$ | $79.21_{\pm0.22}$ | $84.41_{\pm0.32}$ | $76.64_{\pm0.39}$ | $81.94_{\pm0.82}$ |
| **GR-LoRA** | $91.46_{\pm0.11}$ | $94.41_{\pm0.91}$ | $90.03_{\pm0.28}$ | $93.38_{\pm0.96}$ | $80.23_{\pm0.27}$ | $85.05_{\pm0.47}$ | $76.74_{\pm0.63}$ | $82.64_{\pm0.84}$ |

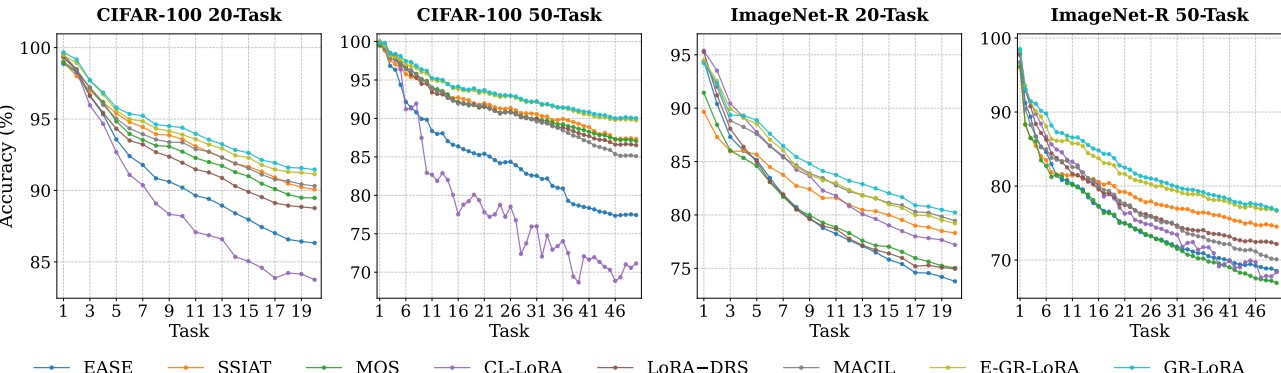

*Figure 3.* The performance of each learning session under different settings of ImageNet-R and CIFAR100. All methods are initialized with ViT-B/16-IN21k. These curves are plotted by calculating the average performance across three different seeds.

As established in Theorem 3.2, by introducing task-specific module to recycle the residual gradient components, the plasticity gap is no longer governed by the entire discarded gradient, but only by the portion that cannot be represented within the subspace spanned by the task-specific module. This result demonstrates that explicitly recycling non-orthogonal gradient components into task-specific module effectively restores the model's usable optimization space, thereby mitigating the plasticity gap induced by orthogonal constraints. The detailed proofs are provided in Appendix C.

## 4. Experiments

### 4.1. Setup

**Datasets.** We train and evaluate our method on four widely used CIL benchmarks. The benchmarks include CIFAR-100 (Krizhevsky et al., 2009) with 100 categories, and ImageNet-R (Hendrycks et al., 2021a), CUB-200 (Wah

et al., 2011), and ImageNet-A (Hendrycks et al., 2021b), each comprising 200 categories. Following standard CIL protocols (Liu & Chang, 2025), all datasets are evaluated under 10-task, 20-task, and 50-task settings.

**Evaluation metrics.** Following CIL protocols (Liu & Chang, 2025), we evaluate performance using two metrics: the final accuracy $\mathcal{A}_{Last}$ and the average accuracy $\mathcal{A}_{Avg}$. Let $a_{i,j}$ denote the test accuracy on the $j$-th task ($j \leq i$) after learning the $i$-th task. The average accuracy at stage $i$ is defined as $\mathcal{A}_i = \frac{1}{i} \sum_{j=1}^{i} a_{i,j}$. Accordingly, the final accuracy is given by $\mathcal{A}_{Last} = \frac{1}{T} \sum_{j=1}^{T} a_{T,j}$, where $T$ denotes the total number of tasks, and the overall average accuracy is computed as $\mathcal{A}_{Avg} = \frac{1}{T} \sum_{i=1}^{T} \mathcal{A}_i$.

**Baselines.** We consider four categories of PTM-based CIL methods for comparison: (1) prompt-based methods, including L2P (Wang et al., 2022b), DualPrompt (Wang et al., 2022a), and CODA-Prompt (Smith et al., 2023); (2)

*Table 2.* Last and average performance results on four benchmark datasets (10 tasks) are reported. The mean and standard deviation of three trials are provided. We compare all methods using the same ViT-B/16-IN21K backbone, seeds, and class orders. Red denotes the best result and blue denotes the second-best result in each column.

| Method | CIFAR-100 | | ImageNet-A | | ImageNet-R | | CUB-200 | |
|---|---|---|---|---|---|---|---|---|
| | $\mathcal{A}_{Last}$ | $\mathcal{A}_{Avg}$ | $\mathcal{A}_{Last}$ | $\mathcal{A}_{Avg}$ | $\mathcal{A}_{Last}$ | $\mathcal{A}_{Avg}$ | $\mathcal{A}_{Last}$ | $\mathcal{A}_{Avg}$ |
| L2P (Wang et al., 2022b) | $84.06_{\pm0.88}$ | $88.26_{\pm1.34}$ | $44.04_{\pm0.93}$ | $51.24_{\pm2.26}$ | $72.34_{\pm0.17}$ | $77.36_{\pm0.64}$ | $67.02_{\pm1.90}$ | $79.62_{\pm1.60}$ |
| DualPrompt (Wang et al., 2022a) | $86.93_{\pm0.24}$ | $91.13_{\pm0.32}$ | $53.19_{\pm0.74}$ | $64.59_{\pm0.08}$ | $69.10_{\pm0.62}$ | $74.28_{\pm0.66}$ | $68.48_{\pm0.47}$ | $80.59_{\pm1.50}$ |
| CODA-Prompt (Smith et al., 2023) | $83.21_{\pm3.39}$ | $87.71_{\pm3.17}$ | $52.08_{\pm0.12}$ | $63.92_{\pm0.12}$ | $73.31_{\pm0.50}$ | $78.47_{\pm0.53}$ | $77.23_{\pm1.12}$ | $81.90_{\pm0.85}$ |
| SLCA (Zhang et al., 2023a) | $91.26_{\pm0.37}$ | $94.29_{\pm0.92}$ | $61.05_{\pm0.63}$ | $68.88_{\pm2.31}$ | $79.35_{\pm0.28}$ | $83.29_{\pm0.46}$ | $84.68_{\pm0.09}$ | $90.77_{\pm0.79}$ |
| InfLoRA (Liang & Li, 2024) | $86.20_{\pm0.70}$ | $90.58_{\pm1.52}$ | $47.75_{\pm0.51}$ | $58.13_{\pm0.56}$ | $75.88_{\pm0.32}$ | $81.90_{\pm0.65}$ | $69.04_{\pm1.25}$ | $81.83_{\pm0.45}$ |
| EASE (Zhou et al., 2024) | $88.22_{\pm0.44}$ | $92.02_{\pm0.76}$ | $54.93_{\pm1.14}$ | $63.92_{\pm0.76}$ | $75.91_{\pm0.17}$ | $81.38_{\pm0.29}$ | $85.04_{\pm1.42}$ | $90.93_{\pm1.03}$ |
| SSIAT (Tan et al., 2024) | $91.48_{\pm0.24}$ | $94.28_{\pm0.90}$ | $62.58_{\pm1.58}$ | $70.73_{\pm1.44}$ | $79.54_{\pm0.24}$ | $83.67_{\pm0.57}$ | $89.83_{\pm0.53}$ | $93.76_{\pm0.52}$ |
| CL-LoRA (He et al., 2025) | $87.47_{\pm0.60}$ | $91.37_{\pm1.30}$ | $57.12_{\pm1.31}$ | $68.17_{\pm1.91}$ | $79.78_{\pm0.17}$ | $85.10_{\pm0.67}$ | $76.28_{\pm2.70}$ | $86.81_{\pm1.24}$ |
| LoRA-DRS (Liu & Chang, 2025) | $90.03_{\pm0.07}$ | $93.24_{\pm0.97}$ | $57.58_{\pm0.79}$ | $66.72_{\pm0.79}$ | $75.96_{\pm0.36}$ | $81.82_{\pm0.85}$ | $87.58_{\pm0.28}$ | $92.30_{\pm0.61}$ |
| MOS (Sun et al., 2025) | $91.54_{\pm0.45}$ | $94.18_{\pm1.15}$ | $57.54_{\pm0.37}$ | $64.50_{\pm1.44}$ | $77.65_{\pm0.50}$ | $81.94_{\pm0.66}$ | $89.88_{\pm0.29}$ | $93.52_{\pm0.61}$ |
| MACIL (Wu et al., 2025) | $91.86_{\pm0.22}$ | $94.44_{\pm0.96}$ | $63.15_{\pm0.17}$ | $70.54_{\pm1.79}$ | $81.82_{\pm0.22}$ | $85.76_{\pm0.32}$ | $90.23_{\pm0.13}$ | $93.78_{\pm0.40}$ |
| **E-GR-LoRA** | $92.00_{\pm0.03}$ | $94.56_{\pm0.96}$ | $63.22_{\pm0.86}$ | $69.96_{\pm2.20}$ | $80.59_{\pm0.21}$ | $85.13_{\pm0.29}$ | $89.79_{\pm0.15}$ | $93.72_{\pm0.51}$ |
| **GR-LoRA** | $91.97_{\pm0.17}$ | $94.65_{\pm0.97}$ | $63.60_{\pm0.41}$ | $70.24_{\pm1.85}$ | $82.09_{\pm0.18}$ | $86.20_{\pm0.28}$ | $89.91_{\pm0.44}$ | $93.85_{\pm0.73}$ |

adapter-based methods, such as SSIAT (Tan et al., 2024), EASE (Zhou et al., 2024), and MOS (Sun et al., 2025); (3) LoRA-based methods, including InfLoRA (Liang & Li, 2024), LoRA-DRS (Liu & Chang, 2025), CL-LoRA (He et al., 2025), and MACIL (Wu et al., 2025); and (4) the fine-tuning method SLCA (Zhang et al., 2023a).

**Implementation Details.** We adopt ViT-B/16 (Dosovitskiy et al., 2021) pretrained on ImageNet-21K (Russakovsky et al., 2015) as the backbone and integrate LoRA with rank $r = 10$ into the key and value projections of all attention layers. Following prior work (Liang & Li, 2024; Liu & Chang, 2025), the model is optimized with Adam and a cosine annealing schedule. All results are reported as the mean and standard deviation over three runs with different random seeds. More implementation details and hyperparameter settings are provided in the Appendix A. The source code is available at https://github.com/njustkmg/ICML26-GR-LoRA.

### 4.2. Main Results

**Performance on Long Task Sequences.** We evaluate the effectiveness of GR-LoRA under long task sequences, with results summarized in Table 1. The experimental results lead to the following observations: (1) GR-LoRA consistently outperforms all competing methods across CIFAR-100 and the more challenging ImageNet-R benchmark under both the 20-task and 50-task settings, achieving superior performance. This demonstrates its strong robustness to error accumulation and distributional drift in long task CIL. (2) Compared with orthogonality-based methods, GR-LoRA attains a more favorable stability–plasticity trade-off by explicitly recycling discarded gradients into task-specific modules, rather than suppressing them through hard constraints. This design allows the model to preserve previously ac-

quired knowledge while maintaining sufficient plasticity for learning new tasks, and the resulting advantage becomes increasingly pronounced as the task sequence length grows. (3) The efficient variant E-GR-LoRA consistently ranks second across all settings, indicating that the proposed gradient recycling remains effective even under reduced computational budgets and confirming its robustness to practical resource constraints. Additional results on long task sequences are provided in Appendix B.1.

**Versatility on Standard Benchmarks.** To assess the generalizability of GR-LoRA, we further evaluate it under standard short-task settings with 10 tasks on four benchmark datasets, as reported in Table 2. The experimental results can be summarized as follows: (1) GR-LoRA demonstrates consistently strong performance across all evaluated settings, indicating robust generalization under short task sequences; (2) on more challenging benchmarks, including ImageNet-A and the fine-grained CUB-200, GR-LoRA remains competitive and maintains stable performance. Detailed per-task results are reported in Appendix B.2.

### 4.3. Ablation Study

**Impact of each component.** As shown in Table 3, we analyze the impact of each proposed component on ImageNet-R. For a fair comparison, following SLCA (Zhang et al., 2023a), we adopt Classifier Alignment applied after training as the baseline, which corresponds to the component in GMS that focuses solely on preserving ID classification performance. The experimental results lead to the following observations: (1) under the presence of CA, recycling the components discarded during orthogonal projection into task-specific modules yields a clear performance improvement, demonstrating the effectiveness of gradient reuse; (2) to mitigate potential inference-time degradation caused by

mis-selection of task-specific module, the two proposed suppression strategies consistently improve model performance, validating their necessity in our method; (3) compared with the "Single LoRA" strategy that allocates an independent adapter for each task, our method achieves significantly better performance, highlighting the importance of sharing orthogonal knowledge across tasks rather than isolating parameters for individual tasks. Additional ablation results on other Datasets are provided in Appendix B.3, further confirming the generalizability of these components across different datasets.

*Table 3.* Ablation study of individual component contributions on the 20-task ImageNet-R benchmark. **CA** denotes Classifier Alignment, **LMS** denotes Local Mismatch Suppression, and **GMS** denotes Global Mismatch Suppression. ✓ and ✗ indicate whether the corresponding component is enabled or disabled, respectively.

| Components
Ablations | CA | LMS | GMS | $\mathcal{A}_{Last}$ | $\mathcal{A}_{Avg}$ |
|---|---|---|---|---|---|
| **LoRA-DRS** | ✓ | ✗ | ✗ | $77.90_{\pm0.26}$ | $82.50_{\pm0.43}$ |
| **Single LoRA** | ✓ | ✓ | ✓ | $70.50_{\pm0.93}$ | $76.79_{\pm0.53}$ |
| **GR-LoRA** | ✓ | ✗ | ✗ | $78.22_{\pm0.44}$ | $83.77_{\pm0.21}$ |
| | ✓ | ✓ | ✗ | $79.12_{\pm0.24}$ | $84.33_{\pm0.22}$ |
| | ✓ | ✗ | ✓ | $79.65_{\pm0.30}$ | $84.73_{\pm0.05}$ |
| | ✓ | ✓ | ✓ | $80.23_{\pm0.27}$ | $85.05_{\pm0.47}$ |

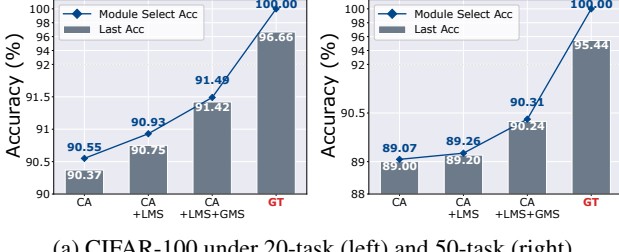

(a) CIFAR-100 under 20-task (left) and 50-task (right).

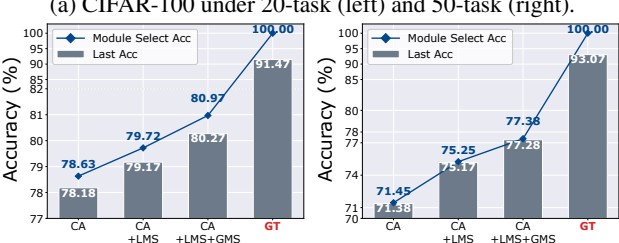

(b) ImagenetR under 20-task (left) and 50-task (right).

*Figure 4.* Module selection accuracy and Last accuracy across different datasets under the 20-task and 50-task setting.

**Module Select Accuracy Analysis.** As shown in Figure 4, we analyze the evolution of module selection accuracy for each task on CIFAR-100 and ImageNet-R under the 20-task and 50-task setting after integrating LMS and GMS. The results show that the proposed suppression strategies significantly improve the final model performance by increasing the accuracy of task-specific module selection at inference

time. Specifically, the two suppression strategies encourage each module to exhibit correct behavior on features corresponding to its associated task, while reducing confidence on mismatched features originating from other tasks. This design is analogous to a $K+1$ classification in OOD detection (Miao et al., 2024), and effectively suppresses erroneous module selection at inference time.

*Table 4.* Comparison on the 20-task ImageNet-R benchmark in terms of Multiply-Accumulate operations (MACs), learnable parameters (LP), and model performance.

| Method | MACs (G) | LP (M) | $\mathcal{A}_{Last}$ | $\mathcal{A}_{Avg}$ |
|---|---|---|---|---|
| L2P | 35.85 | 0.20 | 69.64 | 75.28 |
| CODA-Prompt | 33.73 | 3.99 | 69.96 | 75.34 |
| SLCA | 16.86 | 85.81 | 75.53 | 80.65 |
| LoRA-DRS | 18.31 | 0.38 | 74.96 | 80.66 |
| SSIAT | 17.10 | 1.20 | 78.31 | 82.35 |
| MOS | 16.92 | 0.76 | 75.04 | 80.39 |
| MACIL | 21.51 | 1.19 | 79.46 | 84.25 |
| E-GR-LoRA | 18.27 | 0.38 | 79.21 | 84.41 |
| GR-LoRA | 18.39 | 0.74 | 80.23 | 85.05 |
| GR-LoRA(r=5) | 17.62 | 0.38 | 80.22 | 84.98 |
| GR-LoRA(r=20) | 19.91 | 1.48 | 80.53 | 85.34 |

**Computational Overhead.** As show in Table 4 we compare the Multiply-Accumulate operations computation (Molchanov et al., 2017), trainable parameters, and accuracy metrics ($\mathcal{A}_{Last}$ and $\mathcal{A}_{Avg}$). The results show that GR-LoRA consistently achieves superior accuracy while incurring only modest training and computation overhead, demonstrating a favorable performance–efficiency trade-off. Moreover, the efficient variant E-GR-LoRA substantially reduces the number of trainable parameters with only negligible performance degradation, making it particularly suitable for resource-constrained scenarios. In addition, sensitivity analysis with respect to the rank $r$ shows that the proposed method is largely insensitive to rank variations. Notably, setting $r = 5$ attains training efficiency comparable to E-GR-LoRA while preserving higher accuracy, further highlighting the robustness of the proposed design.

*Table 5.* Inference latency comparison on the 50-task ImageNet-R benchmark.

| Method | Inference Time (s) | | $\mathcal{A}_{Last}$ |
|---|---|---|---|
| | Per Image | Total Test | |
| LoRA-DRS | 1.78 | 15.93 | 72.37 |
| MOS | 2.56 | 400.85 | 66.95 |
| MACIL | 2.01 | 19.04 | 71.13 |
| GR-LoRA | 2.58 | 427.32 | 77.43 |

We further analyze the inference latency of GR-LoRA and

recent SOTA methods on the 50-task ImageNet-R benchmark, as reported in Table 5. Since our task-inference strategy relies on an entropy-based selection mechanism, each input needs to be evaluated by multiple task-specific branches during inference. This design introduces non-negligible time overhead, which increases with the number of tasks. However, such linear computational scaling is a systemic limitation of entropy-based routing architectures rather than a limitation specific to GR-LoRA. Importantly, the accuracy gains brought by GR-LoRA are sufficient to compensate for this additional inference cost.

## 5. Conclusion

In this paper, we propose GR-LoRA to resolve the stability-plasticity dilemma in CIL via a Gradient Recycling mechanism. By utilizing task-specific LoRA modules to recycle non-orthogonal gradient updates and employing entropy-based selection, our method theoretically overcomes the limitations of strict orthogonality while balancing stability and plasticity. To ensure reliable selection, we further propose two suppression strategies consisting of LMS and GMS that minimizes mismatched module activation. Extensive experiments validate the superiority of GR-LoRA, particularly in long-sequence tasks. Additionally, we demonstrate that both our efficient E-GR-LoRA variant and simple rank adjustments offer flexible trade-offs between high parameter efficiency and superior performance. Despite these advantages, the entropy-based selection mechanism introduces additional inference latency as the number of tasks increases. Reducing this latency while preserving task-specific routing benefits remains important future work.

## Acknowledgement

The authors are grateful to the Area Chairs and the anonymous reviewers for their constructive comments. This work is partially supported by the NSFC (62276131), Natural Science Foundation of Jiangsu Province of China under Grant (BK20240081).

## Impact Statement

This paper presents work whose goal is to advance the field of Machine Learning. There are many potential societal consequences of our work, none which we feel must be specifically highlighted here.

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

## A. Implementation Details.

We utilized a single NVIDIA RTX 4090 GPU for all experimental evaluations. For the network architecture, we adopt ViT-B/16 (Dosovitskiy et al., 2021) with $N = 12$ transformer blocks pretrained on ImageNet-21K (Russakovsky et al., 2015) as our backbone architecture across all experiments. Consistent with prior works (Liang & Li, 2024; Liu & Chang, 2025), We set LoRA rank to $r = 10$, and integrate the LoRA architecture into the key and value components of all the attention layers in the transformer. The model was optimized using the Adam optimizer with a learning rate of 0.0005 and a cosine annealing scheduler, with a batch size of 64. Across different datasets: each task is trained for 50 epochs on ImageNet-R and ImageNet-A, and 20 epochs on CIFAR100 and CUB-200. Following standard protocol, we report the mean and standard deviation over three runs with different random seeds. The use of random seeds introduces variability in class order across runs, making the evaluation of model performance more challenging.

## B. Additional Experiments.

### B.1. Performance on Long Task Sequences.

To validate the generalizability of our method across all datasets, particularly in long-sequence scenarios, we conducted additional experiments on the challenging CUB-200 and ImageNet-A datasets under long-task settings (20 and 50 sessions). As shown in Table 6, our method consistently achieves the best performance across all settings, significantly outperforming other SOTA methods. Furthermore, to better demonstrate the superiority of our approach, we report the evolution curves of $\mathcal{A}_{Last}$ throughout the entire training process in Figure 5.

*Table 6.* Last and average performance results on CUB-200 and ImageNet-A under the long-term setting (20 and 50 tasks). The mean and standard deviation of three trials are provided. We compare all methods using the same ViT-B/16-IN21K backbone, seeds, and class orders. Red denotes the best result and blue denotes the second-best result in each column.

| Method | CUB-200 | | | | ImageNet-A | | | |
| --- | --- | --- | --- | --- | --- | --- | --- | --- |
| | 20 | | 50 | | 20 | | 50 | |
| | $\mathcal{A}_{Last}$ | $\mathcal{A}_{Avg}$ | $\mathcal{A}_{Last}$ | $\mathcal{A}_{Avg}$ | $\mathcal{A}_{Last}$ | $\mathcal{A}_{Avg}$ | $\mathcal{A}_{Last}$ | $\mathcal{A}_{Avg}$ |
| CODA-Prompt (Smith et al., 2023) | $66.41_{\pm0.81}$ | $78.10_{\pm1.87}$ | $46.25_{\pm0.68}$ | $63.25_{\pm2.69}$ | $45.40_{\pm1.39}$ | $54.55_{\pm0.86}$ | $30.35_{\pm0.96}$ | $41.53_{\pm0.64}$ |
| SLCA (Zhang et al., 2023a) | $82.48_{\pm0.53}$ | $90.14_{\pm1.02}$ | $78.47_{\pm1.85}$ | $88.38_{\pm0.24}$ | $55.01_{\pm2.66}$ | $63.59_{\pm2.20}$ | $49.31_{\pm0.93}$ | $56.72_{\pm1.82}$ |
| EASE (Zhou et al., 2024) | $84.51_{\pm1.67}$ | $91.02_{\pm1.02}$ | $86.46_{\pm0.14}$ | $92.38_{\pm0.25}$ | $50.32_{\pm2.11}$ | $61.77_{\pm1.60}$ | $37.06_{\pm0.50}$ | $49.80_{\pm0.24}$ |
| SSIAT (Tan et al., 2024) | $89.06_{\pm0.66}$ | $93.52_{\pm0.55}$ | $83.64_{\pm1.67}$ | $91.84_{\pm0.47}$ | $60.52_{\pm2.07}$ | $68.87_{\pm2.29}$ | $51.44_{\pm0.57}$ | $61.63_{\pm2.61}$ |
| LoRA-DRS (Liu & Chang, 2025) | $87.50_{\pm0.26}$ | $92.66_{\pm0.57}$ | $86.97_{\pm0.13}$ | $92.70_{\pm0.30}$ | $55.02_{\pm1.74}$ | $64.52_{\pm1.16}$ | $53.28_{\pm1.75}$ | $63.44_{\pm2.78}$ |
| MOS (Sun et al., 2025) | $89.42_{\pm0.22}$ | $93.66_{\pm0.42}$ | $89.28_{\pm0.25}$ | $93.47_{\pm0.43}$ | $55.26_{\pm0.92}$ | $64.53_{\pm1.02}$ | $52.29_{\pm1.58}$ | $62.91_{\pm1.10}$ |
| MACIL (Wu et al., 2025) | $88.63_{\pm0.61}$ | $93.52_{\pm0.43}$ | $82.06_{\pm0.48}$ | $91.04_{\pm0.41}$ | $59.40_{\pm0.76}$ | $67.79_{\pm1.51}$ | $47.86_{\pm1.78}$ | $59.96_{\pm1.44}$ |
| **GR-LoRA** | $89.76_{\pm0.25}$ | $94.08_{\pm0.55}$ | $89.68_{\pm0.26}$ | $93.94_{\pm0.53}$ | $62.37_{\pm0.34}$ | $69.30_{\pm1.08}$ | $59.71_{\pm0.13}$ | $67.23_{\pm2.16}$ |

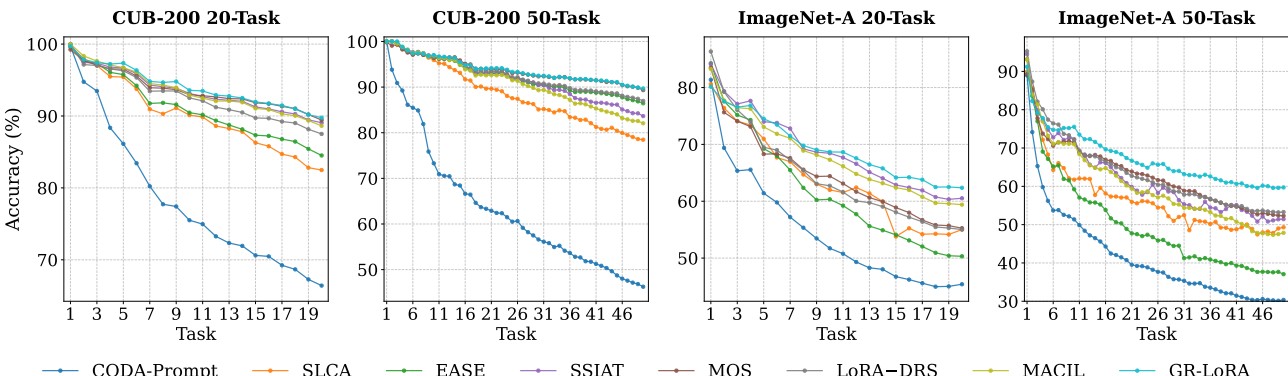

*Figure 5.* The performance of each learning session under different settings of CUB-200 and ImageNet-A. All methods are initialized with ViT-B/16-IN21k. These curves are plotted by calculating the average performance across three different seeds.

## B.2. Versatility on Standard Benchmarks.

To illustrate the performance evolution during the incremental learning process on standard benchmarks, we plot the accuracy curves ($\mathcal{A}_{Last}$) after each task, as shown in Figure 6. It is evident that our method consistently maintains superior performance throughout the entire training sequence, occupying the leading position among all state-of-the-art baselines.

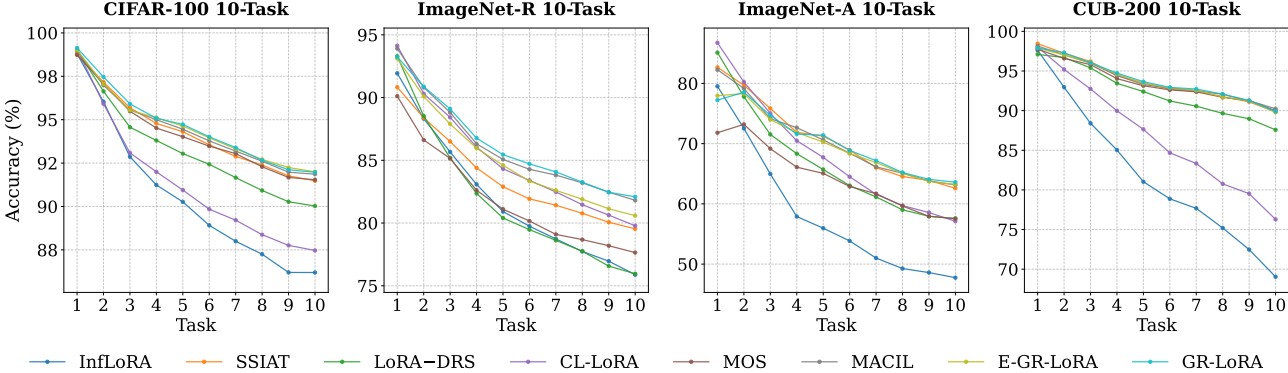

*Figure 6.* The performance of each learning session under four benchmark datasets (10 tasks). All methods are initialized with ViT-B/16-IN21k. These curves are plotted by calculating the average performance across three different seeds.

## B.3. Ablation Study on other Datasets.

Additional ablation studies are conducted on CIFAR-100 and ImageNet-A under the 20-task CIL setting. As shown in Table 7, progressively incorporating the proposed components consistently improves performance across both datasets. These results validate the effectiveness of each module and demonstrate the generalizability of GR-LoRA under diverse and more challenging data distributions.

*Table 7.* Ablation study of individual component contributions on CIFAR-100 and ImageNet-A under the 20-task CIL setting.

| Method | CA | LMS | GMS | CIFAR-100 | | ImageNet-A | |
|---|---|---|---|---|---|---|---|
| | | | | $\mathcal{A}_{Last}$ | $\mathcal{A}_{Avg}$ | $\mathcal{A}_{Last}$ | $\mathcal{A}_{Avg}$ |
| LoRA-DRS | ✓ | × | × | 90.34 | 93.93 | 55.69 | 64.39 |
| Single LoRA | ✓ | ✓ | ✓ | 88.06 | 92.41 | 58.53 | 64.38 |
| GR-LoRA | ✓ | × | × | 90.41 | 94.26 | 58.20 | 66.71 |
| | ✓ | ✓ | × | 90.74 | 94.45 | 59.12 | 67.91 |
| | ✓ | × | ✓ | 91.25 | 94.77 | 61.29 | 70.09 |
| | ✓ | ✓ | ✓ | **91.42** | **94.84** | **62.08** | **70.55** |

## B.4. Storage Overhead.

Storing class prototypes and covariance statistics is a common and effective strategy in CIL, as it helps mitigate the bias of the classifier toward newly learned tasks and improves the retention of previously acquired knowledge. Compared with standard prototype-based classifier or classifier alignment methods, GR-LoRA introduces only negligible additional storage overhead. Specifically, our method requires storing one extra shared-space prototype $\mu_c \in \mathbb{R}^{1 \times 768}$ per class. This additional prototype is used only for constructing OOD prototypes. Importantly, GR-LoRA does not require storing any extra covariance matrix, the generated OOD samples reuse the same private-space covariance matrix $\hat{\Sigma}_c$ as the original ID samples. As shown in Table 8, the storage cost of GR-LoRA is 2.365 MB per class, compared with 2.362 MB for standard Classify Alignment. Therefore, the additional overhead is only about 3 KB per class, which is negligible in practice.

*Table 8.* Storage overhead comparison of representative prototype-based CIL methods per class with feature dimension $d = 768$.

| Method | Stored Variables | Parameters | Storage (MB) |
|---|---|---|---|
| Prototype Classifier | $\mu$ | $d$ | 0.003 |
| Classifier Alignment | $\mu + \Sigma$ | $d + d^2$ | 2.362 |
| GR-LoRA | $\mu + \hat{\mu} + \hat{\Sigma}$ | $2d + d^2$ | 2.365 |

## C. Theoretical proof process.

To analyze the plasticity of the proposed method, we study the attainable population risk under different parameter constraints from an optimization perspective. We assume that the population risk $R_t(\mathbf{W})$ is $\mu$-strongly convex and $L$-smooth with respect to the model parameters $\mathbf{W}$. Let $\mathbf{P}_t^l \in \mathbb{R}^{d \times k}$ denote an orthonormal basis of the residual subspace at layer $l$ for task $t$, i.e., $(\mathbf{P}_t^l)^\top \mathbf{P}_t^l = \mathbf{I}_k$. The corresponding orthogonal projection operator is defined as $\mathbf{\Pi}_t^l = \mathbf{P}_t^l (\mathbf{P}_t^l)^\top$, and the complementary projection is $\mathbf{Q}_t^l = \mathbf{I} - \mathbf{\Pi}_t^l$.

**Theorem C.1.** *For each task $t = 1, \ldots, T$, let $\widehat{\mathbf{W}}_{\perp,t}$ denote the solution obtained under the orthogonal constraint induced by the layer-wise projectors $\{\mathbf{\Pi}_t^l\}_{l=1}^L$, and let $\mathbf{W}_t^\star$ be the unconstrained population minimizer of $R_t$. Then, with probability at least $1 - \delta$, it holds that*

$$\sum_{t=1}^T \left( R_t(\widehat{\mathbf{W}}_{\perp,t}) - R_t(\mathbf{W}_t^\star) \right) \leq \sum_{t=1}^T \sum_{l=1}^L \left( \frac{1}{2\mu} \left\| \mathbf{Q}_t^l \nabla_{\mathbf{W}^l} R_t(\mathbf{W}_{\perp,t}^\star) \right\|_F^2 \right.$$
$$\left. + 4 \mathfrak{R}_{n_t}(\mathcal{F}_{\perp,t}^l) + 2\sqrt{\frac{\log(2TL/\delta)}{2n_t}} \right), \tag{16}$$

*where $\mathbf{W}_{\perp,t}^\star$ is the population minimizer under the orthogonal constraint and $\mathfrak{R}_{n_t}(\cdot)$ denotes the Rademacher complexity.*

*Proof.* For each task $t$, let the data distribution be $\mathcal{P}_t$ over $(\mathbf{x}, y)$. Define the population risk and empirical risk (w.r.t. samples $\mathcal{D}_t = \{(\mathbf{x}_{i,t}, y_{i,t})\}_{i=1}^{n_t}$) as

$$R_t(\mathbf{W}) := \mathbb{E}_{(\mathbf{x},y) \sim \mathcal{P}_t} \left[ \ell(f(\mathbf{x}; \mathbf{W}), y) \right], \qquad \widehat{R}_t(\mathbf{W}) := \frac{1}{n_t} \sum_{i=1}^{n_t} \ell(f(\mathbf{x}_{i,t}; \mathbf{W}), y_{i,t}). \tag{17}$$

Let the orthogonal constraint set be

$$\mathcal{S}_t := \left\{ \mathbf{W} \; : \; \mathbf{W}^l = \mathbf{W}_{t-1}^l + \mathbf{\Pi}_t^l \mathbf{U}^l \text{ for some } \mathbf{U}^l, \; \forall l \in \{1, \ldots, L\} \right\}, \tag{18}$$

i.e., the layer-wise update $\Delta \mathbf{W}^l := \mathbf{W}^l - \mathbf{W}_{t-1}^l$ is restricted to $\text{range}(\mathbf{\Pi}_t^l)$. Define

$$\mathbf{W}_{\perp,t}^\star := \arg\min_{\mathbf{W} \in \mathcal{S}_t} R_t(\mathbf{W}), \qquad \widehat{\mathbf{W}}_{\perp,t} := \arg\min_{\mathbf{W} \in \mathcal{S}_t} \widehat{R}_t(\mathbf{W}), \qquad \mathbf{W}_t^\star := \arg\min_{\mathbf{W}} R_t(\mathbf{W}). \tag{19}$$

Assume $R_t(\mathbf{W})$ is $\mu$-strongly convex and $L$-smooth in $\mathbf{W}$ (Polovinkin, 1996). For each task $t$,

$$R_t(\widehat{\mathbf{W}}_{\perp,t}) - R_t(\mathbf{W}_t^\star) = \underbrace{\left( R_t(\widehat{\mathbf{W}}_{\perp,t}) - R_t(\mathbf{W}_{\perp,t}^\star) \right)}_{\text{estimation / generalization}} + \underbrace{\left( R_t(\mathbf{W}_{\perp,t}^\star) - R_t(\mathbf{W}_t^\star) \right)}_{\text{approximation / constraint-induced}}. \tag{20}$$

Summing (20) over $t = 1, \ldots, T$ reduces the proof to bounding the two terms on the right-hand side. We first state a standard consequence of $\mu$-strong convexity.

**Lemma C.2.** *If a differentiable function $F$ is $\mu$-strongly convex, then for its unique minimizer $\mathbf{W}^\star = \arg\min_{\mathbf{W}} F(\mathbf{W})$,*

$$F(\mathbf{W}) - F(\mathbf{W}^\star) \leq \frac{1}{2\mu} \left\| \nabla F(\mathbf{W}) \right\|_F^2, \qquad \forall \mathbf{W}. \tag{21}$$

*Proof of Lemma C.2.* By $\mu$-strong convexity, for all $\mathbf{U}, \mathbf{V}$,

$$F(\mathbf{U}) \geq F(\mathbf{V}) + \langle \nabla F(\mathbf{V}), \mathbf{U} - \mathbf{V} \rangle + \frac{\mu}{2} \|\mathbf{U} - \mathbf{V}\|_F^2. \tag{22}$$

Set $\mathbf{U} = \mathbf{W}^\star$ and $\mathbf{V} = \mathbf{W}$, rearrange:

$$F(\mathbf{W}) - F(\mathbf{W}^\star) \leq \langle \nabla F(\mathbf{W}), \mathbf{W} - \mathbf{W}^\star \rangle - \frac{\mu}{2} \|\mathbf{W} - \mathbf{W}^\star\|_F^2. \tag{23}$$

Apply Cauchy–Schwarz and the inequality $ab - \frac{\mu}{2}b^2 \leq \frac{1}{2\mu}a^2$ with $a = \|\nabla F(\mathbf{W})\|_F$ and $b = \|\mathbf{W} - \mathbf{W}^\star\|_F$:

$$\langle \nabla F(\mathbf{W}), \mathbf{W} - \mathbf{W}^\star \rangle - \frac{\mu}{2} \|\mathbf{W} - \mathbf{W}^\star\|_F^2 \leq \|\nabla F(\mathbf{W})\|_F \|\mathbf{W} - \mathbf{W}^\star\|_F - \frac{\mu}{2} \|\mathbf{W} - \mathbf{W}^\star\|_F^2 \leq \frac{1}{2\mu} \|\nabla F(\mathbf{W})\|_F^2. \tag{24}$$

This yields (21). $\qquad\square$

We now bound $R_t(\mathbf{W}_{\perp,t}^\star) - R_t(\mathbf{W}_t^\star)$. Because $\mathcal{S}_t$ is an affine subspace whose feasible directions at layer $l$ equal $\mathrm{range}(\mathbf{\Pi}_t^l)$, the first-order optimality condition for the constrained minimizer $\mathbf{W}_{\perp,t}^\star$ can be written as: for every layer $l$ and for every feasible perturbation $\Delta \mathbf{W}^l \in \mathrm{range}(\mathbf{\Pi}_t^l)$,

$$\langle \nabla_{\mathbf{W}^l} R_t(\mathbf{W}_{\perp,t}^\star), \ \Delta \mathbf{W}^l \rangle = 0. \tag{25}$$

Since $\mathrm{range}(\mathbf{\Pi}_t^l)$ is exactly the set $\{\mathbf{\Pi}_t^l \mathbf{U}^l : \mathbf{U}^l\}$, (25) is equivalent to

$$\langle \nabla_{\mathbf{W}^l} R_t(\mathbf{W}_{\perp,t}^\star), \ \mathbf{\Pi}_t^l \mathbf{U}^l \rangle = 0, \quad \forall \mathbf{U}^l. \tag{26}$$

Using $\langle \mathbf{A}, \mathbf{B} \rangle = \mathrm{tr}(\mathbf{A}^\top \mathbf{B})$ and cyclicity of trace,

$$0 = \mathrm{tr}\big( (\nabla_{\mathbf{W}^l} R_t(\mathbf{W}_{\perp,t}^\star))^\top \mathbf{\Pi}_t^l \mathbf{U}^l \big) = \mathrm{tr}\big( ((\mathbf{\Pi}_t^l)^\top \nabla_{\mathbf{W}^l} R_t(\mathbf{W}_{\perp,t}^\star))^\top \mathbf{U}^l \big), \quad \forall \mathbf{U}^l. \tag{27}$$

Hence,

$$(\mathbf{\Pi}_t^l)^\top \nabla_{\mathbf{W}^l} R_t(\mathbf{W}_{\perp,t}^\star) = \mathbf{0}. \tag{28}$$

Because $\mathbf{\Pi}_t^l$ is an orthogonal projector, it is symmetric: $(\mathbf{\Pi}_t^l)^\top = \mathbf{\Pi}_t^l$. Therefore,

$$\mathbf{\Pi}_t^l \nabla_{\mathbf{W}^l} R_t(\mathbf{W}_{\perp,t}^\star) = \mathbf{0}, \qquad \Longrightarrow \qquad \nabla_{\mathbf{W}^l} R_t(\mathbf{W}_{\perp,t}^\star) = (\mathbf{I} - \mathbf{\Pi}_t^l) \nabla_{\mathbf{W}^l} R_t(\mathbf{W}_{\perp,t}^\star) = \mathbf{Q}_t^l \nabla_{\mathbf{W}^l} R_t(\mathbf{W}_{\perp,t}^\star). \tag{29}$$

Now apply Lemma C.2 to $F(\mathbf{W}) = R_t(\mathbf{W})$ at $\mathbf{W} = \mathbf{W}_{\perp,t}^\star$:

$$R_t(\mathbf{W}_{\perp,t}^\star) - R_t(\mathbf{W}_t^\star) \leq \frac{1}{2\mu} \big\| \nabla R_t(\mathbf{W}_{\perp,t}^\star) \big\|_F^2. \tag{30}$$

Finally, expand the full gradient norm layer-wise and use (29):

$$\big\| \nabla R_t(\mathbf{W}_{\perp,t}^\star) \big\|_F^2 = \sum_{l=1}^L \big\| \nabla_{\mathbf{W}^l} R_t(\mathbf{W}_{\perp,t}^\star) \big\|_F^2 = \sum_{l=1}^L \big\| \mathbf{Q}_t^l \nabla_{\mathbf{W}^l} R_t(\mathbf{W}_{\perp,t}^\star) \big\|_F^2. \tag{31}$$

Thus we obtain the constraint-induced gap:

$$R_t(\mathbf{W}_{\perp,t}^\star) - R_t(\mathbf{W}_t^\star) \leq \sum_{l=1}^L \frac{1}{2\mu} \big\| \mathbf{Q}_t^l \nabla_{\mathbf{W}^l} R_t(\mathbf{W}_{\perp,t}^\star) \big\|_F^2. \tag{32}$$

We now bound $R_t(\widehat{\mathbf{W}}_{\perp,t}) - R_t(\mathbf{W}_{\perp,t}^\star)$. Let $\mathcal{F}_{\perp,t}$ be the hypothesis class induced by parameters in $\mathcal{S}_t$:

$$\mathcal{F}_{\perp,t} := \{ (\mathbf{x} \mapsto \ell(f(\mathbf{x}; \mathbf{W}), y)) : \mathbf{W} \in \mathcal{S}_t \}. \tag{33}$$

Assume we have an additive upper bound on the Rademacher complexity:

$$\mathfrak{R}_{n_t}(\mathcal{F}_{\perp,t}) \leq \sum_{l=1}^{L} \mathfrak{R}_{n_t}(\mathcal{F}_{\perp,t}^l). \tag{34}$$

We use the standard uniform convergence bound: for any $\delta_t \in (0,1)$, with probability at least $1 - \delta_t$,

$$\sup_{f \in \mathcal{F}_{\perp,t}} \left( R_t(f) - \widehat{R}_t(f) \right) \leq 2\,\mathfrak{R}_{n_t}(\mathcal{F}_{\perp,t}) + \sqrt{\frac{\log(1/\delta_t)}{2n_t}}. \tag{35}$$

Similarly, with probability at least $1 - \delta_t$,

$$\sup_{f \in \mathcal{F}_{\perp,t}} \left( \widehat{R}_t(f) - R_t(f) \right) \leq 2\,\mathfrak{R}_{n_t}(\mathcal{F}_{\perp,t}) + \sqrt{\frac{\log(1/\delta_t)}{2n_t}}. \tag{36}$$

By a union bound over (35) and (36), with probability at least $1 - 2\delta_t$,

$$\sup_{f \in \mathcal{F}_{\perp,t}} \left| R_t(f) - \widehat{R}_t(f) \right| \leq 2\,\mathfrak{R}_{n_t}(\mathcal{F}_{\perp,t}) + \sqrt{\frac{\log(1/\delta_t)}{2n_t}}. \tag{37}$$

Now use the ERM property:

$$\widehat{R}_t(\widehat{\mathbf{W}}_{\perp,t}) \leq \widehat{R}_t(\mathbf{W}_{\perp,t}^\star). \tag{38}$$

Then, on the event (37),

$$
\begin{aligned}
R_t(\widehat{\mathbf{W}}_{\perp,t}) - R_t(\mathbf{W}_{\perp,t}^\star) &= \left( R_t(\widehat{\mathbf{W}}_{\perp,t}) - \widehat{R}_t(\widehat{\mathbf{W}}_{\perp,t}) \right) + \left( \widehat{R}_t(\widehat{\mathbf{W}}_{\perp,t}) - \widehat{R}_t(\mathbf{W}_{\perp,t}^\star) \right) + \left( \widehat{R}_t(\mathbf{W}_{\perp,t}^\star) - R_t(\mathbf{W}_{\perp,t}^\star) \right) \\
&\leq \left| R_t(\widehat{\mathbf{W}}_{\perp,t}) - \widehat{R}_t(\widehat{\mathbf{W}}_{\perp,t}) \right| + 0 + \left| \widehat{R}_t(\mathbf{W}_{\perp,t}^\star) - R_t(\mathbf{W}_{\perp,t}^\star) \right| \\
&\leq 2 \sup_{\mathbf{W} \in \mathcal{S}_t} \left| R_t(\mathbf{W}) - \widehat{R}_t(\mathbf{W}) \right| \\
&\leq 4\,\mathfrak{R}_{n_t}(\mathcal{F}_{\perp,t}) + 2\sqrt{\frac{\log(1/\delta_t)}{2n_t}}.
\end{aligned}
\tag{39}
$$

Combining with (34), we further obtain

$$R_t(\widehat{\mathbf{W}}_{\perp,t}) - R_t(\mathbf{W}_{\perp,t}^\star) \leq \sum_{l=1}^{L} \left( 4\,\mathfrak{R}_{n_t}(\mathcal{F}_{\perp,t}^l) + 2\sqrt{\frac{\log(1/\delta_t)}{2n_t}} \right). \tag{40}$$

Choose $\delta_t$ so that the total failure probability across all tasks and layers is at most $\delta$. A convenient choice is $\delta_t := \frac{\delta}{TL}$, and we already paid a factor 2 in (37), hence the logarithmic term becomes $\log(2TL/\delta)$. Substituting $\delta_t = \delta/(TL)$ into (40) yields, simultaneously for all $t$ (and all layers accounted in the sum),

$$R_t(\widehat{\mathbf{W}}_{\perp,t}) - R_t(\mathbf{W}_{\perp,t}^\star) \leq \sum_{l=1}^{L} \left( 4\,\mathfrak{R}_{n_t}(\mathcal{F}_{\perp,t}^l) + 2\sqrt{\frac{\log(2TL/\delta)}{2n_t}} \right), \tag{41}$$

with probability at least $1 - \delta$ after union bounding over $t = 1, \ldots, T$. Finally, plug (32) and (41) into (20), and sum over $t = 1, \ldots, T$:

$$
\begin{aligned}
\sum_{t=1}^{T} \left( R_t(\widehat{\mathbf{W}}_{\perp,t}) - R_t(\mathbf{W}_t^\star) \right) &\leq \sum_{t=1}^{T} \left( R_t(\widehat{\mathbf{W}}_{\perp,t}) - R_t(\mathbf{W}_{\perp,t}^\star) + R_t(\mathbf{W}_{\perp,t}^\star) - R_t(\mathbf{W}_t^\star) \right) \\
&\leq \sum_{t=1}^{T} \sum_{l=1}^{L} \left( 4\,\mathfrak{R}_{n_t}(\mathcal{F}_{\perp,t}^l) + 2\sqrt{\frac{\log(2TL/\delta)}{2n_t}} + \frac{1}{2\mu} \left\| \mathbf{Q}_t^l \nabla_{\mathbf{W}^l} R_t(\mathbf{W}_{\perp,t}^\star) \right\|_F^2 \right).
\end{aligned}
\tag{42}
$$

This is exactly the desired bound. $\qquad \square$

As established in Theorem C.1, when model updates are restricted to the orthogonal subspace defined by $\mathbf{\Pi}_t^l$, the cumulative performance gap relative to the unconstrained optimum is dominated by $\|\mathbf{Q}_t^l \nabla R_t\|_F^2$. Consequently, even under perfect samples and optimization, orthogonal constraints induce an unavoidable plasticity gap.

**Theorem C.3.** *Suppose the model parameters are augmented in the LoRA form* $\mathbf{W}^l = \mathbf{W}_0^l + \sum_{j=1}^{t-1} \mathbf{B}_j^l \mathbf{A}_j^l + (\mathbf{B}_t^l + \hat{\mathbf{B}}_t^l)(\mathbf{A}_t^l + \hat{\mathbf{A}}_t^l)$, *where* $\mathbf{B}_t^l \mathbf{A}_t^l$ *denotes the shared module learned under orthogonal constraints, and* $\hat{\mathbf{B}}_t^l \hat{\mathbf{A}}_t^l$ *is a task-specific module allocated to task* $t$. *Let* $\mathbf{W}_{\mathrm{GR},t}^\star$ *denote the population minimizer under this augmented parameterization. Then the cumulative optimality gap across tasks satisfies*

$$\sum_{t=1}^T \Big( R_t(\mathbf{W}_{\mathrm{GR},t}^\star) - R_t(\mathbf{W}_t^\star) \Big) \leq \sum_{t=1}^T \sum_{l=1}^L \frac{1}{2\mu} \big\| \mathbf{Q}_t^l \nabla_{\mathbf{W}^l} R_t(\mathbf{W}_{\perp,t}^\star) - \Pi_{\mathcal{S}_t^l} \big( \mathbf{Q}_t^l \nabla_{\mathbf{W}^l} R_t(\mathbf{W}_{\perp,t}^\star) \big) \big\|_F^2, \qquad (43)$$

*where* $\Pi_{\mathcal{S}_t^l}$ *denotes the projection onto the subspace* $\mathcal{S}_t^l = \mathrm{range}(\hat{\mathbf{B}}_t^l \hat{\mathbf{A}}_t^l)$.

*Proof.* For each layer $l$, define $\mathbf{w}^l := \mathrm{vec}(\mathbf{W}^l) \in \mathbb{R}^{d_l}$ and stack all layers as $\mathbf{w} := ((\mathbf{w}^1)^\top, \ldots, (\mathbf{w}^L)^\top)^\top \in \mathbb{R}^d$, where $d = \sum_{l=1}^L d_l$. We use the standard inner product $\langle \mathbf{a}, \mathbf{b} \rangle = \mathbf{a}^\top \mathbf{b}$ and its matrix form $\langle \mathbf{A}, \mathbf{B} \rangle = \mathrm{tr}(\mathbf{A}^\top \mathbf{B})$. For any operator $\mathbf{M}$, denote its orthogonal projector by the same symbol. Under the quadratic assumption, for each task $t$ there exists a symmetric positive definite matrix $\mathbf{H}_t \in \mathbb{R}^{d \times d}$ such that

$$R_t(\mathbf{w}) = R_t(\mathbf{w}_t^\star) + \frac{1}{2}(\mathbf{w} - \mathbf{w}_t^\star)^\top \mathbf{H}_t(\mathbf{w} - \mathbf{w}_t^\star), \qquad \nabla R_t(\mathbf{w}) = \mathbf{H}_t(\mathbf{w} - \mathbf{w}_t^\star), \qquad (44)$$

with $\mu \mathbf{I} \preceq \mathbf{H}_t \preceq L \mathbf{I}$. In particular, $\mathbf{w}_t^\star$ is the unique unconstrained minimizer and $\nabla R_t(\mathbf{w}_t^\star) = \mathbf{0}$. Let $\mathcal{T}_{\perp,t}$ denote the tangent subspace induced by the layer-wise orthogonal projectors $\{\mathbf{\Pi}_t^l\}_{l=1}^L$:

$$\mathcal{T}_{\perp,t} := \Big\{ \Delta \mathbf{w} = ((\Delta \mathbf{w}^1)^\top, \ldots, (\Delta \mathbf{w}^L)^\top)^\top : \Delta \mathbf{w}^l \in \mathrm{range}(\mathbf{\Pi}_t^l), \forall l \Big\}. \qquad (45)$$

Define the corresponding block-diagonal projector $\mathbf{\Pi}_{\perp,t}$ onto $\mathcal{T}_{\perp,t}$ and $\mathbf{Q}_{\perp,t} := \mathbf{I} - \mathbf{\Pi}_{\perp,t}$. By construction, on each layer block $l$, $(\mathbf{\Pi}_{\perp,t})|_l = \mathbf{\Pi}_t^l$ and $(\mathbf{Q}_{\perp,t})|_l = \mathbf{Q}_t^l$. Let $\mathbf{w}_{\perp,t}^\star$ be the constrained minimizer:

$$\mathbf{w}_{\perp,t}^\star := \arg\min_{\mathbf{w} \in \mathbf{w}_{t-1} + \mathcal{T}_{\perp,t}} R_t(\mathbf{w}). \qquad (46)$$

Since the feasible set is an affine subspace and $R_t$ is differentiable and strongly convex, $\mathbf{w}_{\perp,t}^\star$ is unique and satisfies the first-order optimality condition:

$$\mathbf{\Pi}_{\perp,t} \nabla R_t(\mathbf{w}_{\perp,t}^\star) = \mathbf{0} \qquad \Longleftrightarrow \qquad \nabla R_t(\mathbf{w}_{\perp,t}^\star) = \mathbf{Q}_{\perp,t} \nabla R_t(\mathbf{w}_{\perp,t}^\star). \qquad (47)$$

Define the *discarded gradient* at $\mathbf{w}_{\perp,t}^\star$:

$$\mathbf{g}_t := \mathbf{Q}_{\perp,t} \nabla R_t(\mathbf{w}_{\perp,t}^\star) = \nabla R_t(\mathbf{w}_{\perp,t}^\star), \qquad \mathbf{g}_t^l := \mathbf{Q}_t^l \nabla_{\mathbf{w}^l} R_t(\mathbf{w}_{\perp,t}^\star), \qquad (48)$$

so that $\|\mathbf{g}_t\|_2^2 = \sum_{l=1}^L \|\mathbf{g}_t^l\|_2^2$. Under the GR-LoRA, the update directions include: (i) the orthogonal directions $\mathcal{T}_{\perp,t}$ (shared orthogonal module), and (ii) additional task-specific directions in each layer $l$ belonging to the subspace

$$\mathcal{S}_t^l := \mathrm{range}(\hat{\mathbf{B}}_t^l \hat{\mathbf{A}}_t^l) \subseteq \mathrm{range}(\mathbf{Q}_t^l), \qquad (49)$$

where the inclusion in (49) is the standard GR-LoRA design assumption (task-specific module spans residual directions). Let $\mathcal{T}_{\mathrm{GR},t}$ denote the expanded tangent space: $\mathcal{T}_{\mathrm{GR},t} := \Big\{ \Delta \mathbf{w} : \Delta \mathbf{w}^l \in \mathrm{range}(\mathbf{\Pi}_t^l) \oplus \mathcal{S}_t^l, \forall l \Big\}$. Let $\mathbf{\Pi}_{\mathrm{GR},t}$ be the orthogonal projector onto $\mathcal{T}_{\mathrm{GR},t}$, and $\mathbf{Q}_{\mathrm{GR},t} := \mathbf{I} - \mathbf{\Pi}_{\mathrm{GR},t}$. Let $\mathbf{w}_{\mathrm{GR},t}^\star$ be the population minimizer over the GR feasible affine space:

$$\mathbf{w}_{\mathrm{GR},t}^\star := \arg\min_{\mathbf{w} \in \mathbf{w}_{t-1} + \mathcal{T}_{\mathrm{GR},t}} R_t(\mathbf{w}). \qquad (50)$$

Again, uniqueness holds and the first-order optimality condition yields

$$\mathbf{\Pi}_{\mathrm{GR},t} \nabla R_t(\mathbf{w}_{\mathrm{GR},t}^\star) = \mathbf{0} \qquad \Longleftrightarrow \qquad \nabla R_t(\mathbf{w}_{\mathrm{GR},t}^\star) = \mathbf{Q}_{\mathrm{GR},t} \nabla R_t(\mathbf{w}_{\mathrm{GR},t}^\star). \qquad (51)$$

For a quadratic $R_t$ of the form (44), minimizing $R_t$ over an affine subspace $\mathbf{w}_0 + \mathcal{T}$ is equivalent to projecting the error $\mathbf{w}_t^\star - \mathbf{w}_0$ in the $\mathbf{H}_t$-geometry. A standard characterization is:

$$\mathbf{w}_\mathcal{T}^\star = \arg \min_{\mathbf{w} \in \mathbf{w}_0 + \mathcal{T}} R_t(\mathbf{w}) \quad \Longleftrightarrow \quad \mathbf{w}_\mathcal{T}^\star = \mathbf{w}_0 + \mathbf{\Pi}_\mathcal{T}^{(\mathbf{H}_t)}(\mathbf{w}_t^\star - \mathbf{w}_0), \tag{52}$$

where $\mathbf{\Pi}_\mathcal{T}^{(\mathbf{H}_t)}$ denotes the $\mathbf{H}_t$-orthogonal projection onto $\mathcal{T}$ (i.e., projection under the inner product $\langle \mathbf{a}, \mathbf{b} \rangle_{\mathbf{H}_t} = \mathbf{a}^\top \mathbf{H}_t \mathbf{b}$). Moreover, the optimality gap admits the exact expression

$$R_t(\mathbf{w}_\mathcal{T}^\star) - R_t(\mathbf{w}_t^\star) = \frac{1}{2} \left\| \mathbf{Q}_\mathcal{T}^{(\mathbf{H}_t)}(\mathbf{w}_0 - \mathbf{w}_t^\star) \right\|_{\mathbf{H}_t}^2, \tag{53}$$

where $\mathbf{Q}_\mathcal{T}^{(\mathbf{H}_t)} = \mathbf{I} - \mathbf{\Pi}_\mathcal{T}^{(\mathbf{H}_t)}$ and $\|\mathbf{v}\|_{\mathbf{H}_t}^2 := \mathbf{v}^\top \mathbf{H}_t \mathbf{v}$. We will use a *Euclidean* upper bound of (53) via the spectral lower bound $\mathbf{H}_t \succeq \mu \mathbf{I}$:

$$\|\mathbf{v}\|_{\mathbf{H}_t}^2 \leq \frac{1}{\mu} \|\mathbf{H}_t \mathbf{v}\|_2^2 \quad \text{since} \quad \mathbf{v}^\top \mathbf{H}_t \mathbf{v} \leq \frac{1}{\mu} \mathbf{v}^\top \mathbf{H}_t^2 \mathbf{v} = \frac{1}{\mu} \|\mathbf{H}_t \mathbf{v}\|_2^2. \tag{54}$$

Apply (53) to $\mathcal{T} = \mathcal{T}_{\mathrm{GR},t}$ and $\mathbf{w}_0 = \mathbf{w}_{t-1}$:

$$R_t(\mathbf{w}_{\mathrm{GR},t}^\star) - R_t(\mathbf{w}_t^\star) = \frac{1}{2} \left\| \mathbf{Q}_{\mathrm{GR},t}^{(\mathbf{H}_t)}(\mathbf{w}_{t-1} - \mathbf{w}_t^\star) \right\|_{\mathbf{H}_t}^2. \tag{55}$$

Now define the constrained-only solution $\mathbf{w}_{\perp,t}^\star$ which corresponds to tangent space $\mathcal{T}_{\perp,t}$. By the same formula,

$$R_t(\mathbf{w}_{\perp,t}^\star) - R_t(\mathbf{w}_t^\star) = \frac{1}{2} \left\| \mathbf{Q}_{\perp,t}^{(\mathbf{H}_t)}(\mathbf{w}_{t-1} - \mathbf{w}_t^\star) \right\|_{\mathbf{H}_t}^2. \tag{56}$$

Crucially, the *residual gradient* at $\mathbf{w}_{\perp,t}^\star$ satisfies

$$\mathbf{g}_t = \nabla R_t(\mathbf{w}_{\perp,t}^\star) = \mathbf{H}_t(\mathbf{w}_{\perp,t}^\star - \mathbf{w}_t^\star). \tag{57}$$

Because $\mathbf{w}_{\perp,t}^\star \in \mathbf{w}_{t-1} + \mathcal{T}_{\perp,t}$, we can write

$$\mathbf{w}_{\perp,t}^\star - \mathbf{w}_t^\star = (\mathbf{w}_{t-1} - \mathbf{w}_t^\star) + \Delta_{\perp,t} \quad \text{for some} \quad \Delta_{\perp,t} \in \mathcal{T}_{\perp,t}. \tag{58}$$

Projecting onto the orthogonal complement of $\mathcal{T}_{\mathrm{GR},t}$ eliminates the $\mathcal{T}_{\perp,t}$ component since $\mathcal{T}_{\perp,t} \subseteq \mathcal{T}_{\mathrm{GR},t}$:

$$\mathbf{Q}_{\mathrm{GR},t}(\mathbf{w}_{\perp,t}^\star - \mathbf{w}_t^\star) = \mathbf{Q}_{\mathrm{GR},t}(\mathbf{w}_{t-1} - \mathbf{w}_t^\star). \tag{59}$$

Now use (55) and the bound (54) with $\mathbf{v} = \mathbf{Q}_{\mathrm{GR},t}(\mathbf{w}_{t-1} - \mathbf{w}_t^\star)$:

$$\begin{aligned} R_t(\mathbf{w}_{\mathrm{GR},t}^\star) - R_t(\mathbf{w}_t^\star) = \frac{1}{2} \left\| \mathbf{Q}_{\mathrm{GR},t}^{(\mathbf{H}_t)}(\mathbf{w}_{t-1} - \mathbf{w}_t^\star) \right\|_{\mathbf{H}_t}^2 &\leq \frac{1}{2} \left\| \mathbf{Q}_{\mathrm{GR},t}(\mathbf{w}_{t-1} - \mathbf{w}_t^\star) \right\|_{\mathbf{H}_t}^2 \\ &\leq \frac{1}{2\mu} \left\| \mathbf{H}_t \, \mathbf{Q}_{\mathrm{GR},t}(\mathbf{w}_{t-1} - \mathbf{w}_t^\star) \right\|_2^2. \end{aligned} \tag{60}$$

Using (59) and (57),

$$\mathbf{H}_t \, \mathbf{Q}_{\mathrm{GR},t}(\mathbf{w}_{t-1} - \mathbf{w}_t^\star) = \mathbf{H}_t \, \mathbf{Q}_{\mathrm{GR},t}(\mathbf{w}_{\perp,t}^\star - \mathbf{w}_t^\star) = \mathbf{Q}_{\mathrm{GR},t} \mathbf{H}_t(\mathbf{w}_{\perp,t}^\star - \mathbf{w}_t^\star) = \mathbf{Q}_{\mathrm{GR},t} \, \mathbf{g}_t, \tag{61}$$

where the last equality uses that $\mathbf{Q}_{\mathrm{GR},t}$ is an *Euclidean* orthogonal projector and $\mathbf{g}_t$ is a vector; thus $\mathbf{Q}_{\mathrm{GR},t} \mathbf{g}_t$ is well-defined. Plugging (61) into (60) yields

$$R_t(\mathbf{w}_{\mathrm{GR},t}^\star) - R_t(\mathbf{w}_t^\star) \leq \frac{1}{2\mu} \left\| \mathbf{Q}_{\mathrm{GR},t} \mathbf{g}_t \right\|_2^2. \tag{62}$$

By construction, $\mathcal{T}_{\mathrm{GR},t}$ expands $\mathcal{T}_{\perp,t}$ by adding $\mathcal{S}_t^l$ in each layer. Under the design assumption $\mathcal{S}_t^l \subseteq \mathrm{range}(\mathbf{Q}_t^l)$ in (49), we have an *orthogonal direct sum* per layer:

$$\mathrm{range}(\mathbf{\Pi}_t^l) \perp \mathcal{S}_t^l, \qquad \mathrm{range}(\mathbf{\Pi}_t^l) \oplus \mathcal{S}_t^l \subseteq \mathbb{R}^{d_l}. \tag{63}$$

Hence, the Euclidean orthogonal projector onto $\mathrm{range}(\mathbf{\Pi}_t^l) \oplus \mathcal{S}_t^l$ is

$$\mathbf{\Pi}_{\mathrm{GR},t}^l = \mathbf{\Pi}_t^l + \Pi_{\mathcal{S}_t^l}, \qquad \mathbf{Q}_{\mathrm{GR},t}^l = \mathbf{I} - \mathbf{\Pi}_{\mathrm{GR},t}^l = \mathbf{I} - \mathbf{\Pi}_t^l - \Pi_{\mathcal{S}_t^l}. \tag{64}$$

Now recall from (48) that $\mathbf{g}_t^l \in \mathrm{range}(\mathbf{Q}_t^l)$, i.e., $\mathbf{\Pi}_t^l \mathbf{g}_t^l = \mathbf{0}$. Therefore,

$$\mathbf{Q}_{\mathrm{GR},t}^l \mathbf{g}_t^l = (\mathbf{I} - \mathbf{\Pi}_t^l - \Pi_{\mathcal{S}_t^l}) \mathbf{g}_t^l = \mathbf{g}_t^l - \Pi_{\mathcal{S}_t^l}(\mathbf{g}_t^l). \tag{65}$$

Stacking all layers,

$$\left\|\mathbf{Q}_{\mathrm{GR},t} \mathbf{g}_t\right\|_2^2 = \sum_{l=1}^L \left\|\mathbf{Q}_{\mathrm{GR},t}^l \mathbf{g}_t^l\right\|_2^2 = \sum_{l=1}^L \left\|\mathbf{g}_t^l - \Pi_{\mathcal{S}_t^l}(\mathbf{g}_t^l)\right\|_2^2. \tag{66}$$

Returning to matrix notation, $\|\cdot\|_2$ on $\mathrm{vec}(\cdot)$ equals the Frobenius norm $\|\cdot\|_F$ on matrices, so

$$\left\|\mathbf{g}_t^l - \Pi_{\mathcal{S}_t^l}(\mathbf{g}_t^l)\right\|_2^2 = \left\|\mathbf{Q}_t^l \nabla_{\mathbf{W}^l} R_t(\mathbf{W}_{\perp,t}^\star) - \Pi_{\mathcal{S}_t^l}\!\left(\mathbf{Q}_t^l \nabla_{\mathbf{W}^l} R_t(\mathbf{W}_{\perp,t}^\star)\right)\right\|_F^2. \tag{67}$$

Combine (62) and (66):

$$R_t(\mathbf{W}_{\mathrm{GR},t}^\star) - R_t(\mathbf{W}_t^\star) \le \sum_{l=1}^L \frac{1}{2\mu} \left\|\mathbf{Q}_t^l \nabla_{\mathbf{W}^l} R_t(\mathbf{W}_{\perp,t}^\star) - \Pi_{\mathcal{S}_t^l}\!\left(\mathbf{Q}_t^l \nabla_{\mathbf{W}^l} R_t(\mathbf{W}_{\perp,t}^\star)\right)\right\|_F^2. \tag{68}$$

Summing the above inequality over $t = 1, \dots, T$ yields the claimed result. $\qquad \square$

As established in Theorem C.3, by introducing task-specific low-rank modules to recycle residual gradient components, the plasticity gap is no longer governed by the entire discarded gradient $\mathbf{Q}_t^l \nabla R_t$, but only by the portion that lies outside the representational subspace $\mathcal{S}_t^l$. This result demonstrates that Gradient Recycling effectively restores the usable optimization geometry under orthogonal constraints, thereby mitigating the intrinsic plasticity gap induced by subspace projection.

