# OpenReview forum: "GR-LoRA: Gradient-Recycling Low-Rank Adaptation for Class-Incremental Learning"
_ICML.cc/2026/Conference — ICML 2026 spotlight_

### Official Review · Reviewer_jt37 · 2026-03-04

**Soundness:** 3
**Presentation:** 3
**Significance:** 3
**Originality:** 3
**Overall Recommendation:** 4
**Confidence:** 4

**Summary:**

This paper proposes Gradient-Recycling Low-Rank Adaptation (GR-LoRA) for Class-Incremental Learning (CIL). The central idea is a gradient recycling mechanism that redirects non-orthogonal gradient components, which would otherwise be discarded, into lightweight task-specific LoRA modules. This approach aims to balance stability and plasticity in CIL without suffering from catastrophic forgetting. Experimental results on multiple benchmark datasets demonstrate the effectiveness of the proposed method.

**Compliance With Llm Reviewing Policy:**

Affirmed.

**Final Justification:**

Please refer to the Rebuttal Acknowledgement.

**Key Questions For Authors:**

Key Questions are given in the Strengths and Weaknesses part.

**Limitations:**

This paper does not explicitly discuss the limitations in a separate section.

**Strengths And Weaknesses:**

**Strengths**

1.	The paper is well-motivated, and the proposed method is both innovative and promising.
2.	The theoretical analysis appears reasonable and comprehensive.
3.	The overall presentation is clear, well-organized, and easy to follow.

**Weaknesses**

1.	Although the theoretical analysis is solid, it is still insufficient to fully explain the proposed gradient recycling mechanism. The authors are encouraged to provide more intuitive empirical evidence, such as visualizations and statistical analyses, to better illustrate its behavior.
2.	The proposed system appears complex. A more detailed efficiency analysis of each key component, including memory consumption and inference latency, would be valuable. Although MACs and the number of parameters are reported, they cannot fully reflect real-time computational performance.
3.	The paper does not analyze whether recycling non-orthogonal gradients could introduce the risk of catastrophic forgetting in cases where tasks/classes have high semantic overlap. In such situations, the recycled gradients may contain information related to previous tasks, potentially impairing stability. For instance, on the CUB-200 dataset, the proposed method shows only marginal improvements. This may suggest that GR-LoRA lacks robustness in fine-grained recognition scenarios.
4.	The method is not evaluated on longer task sequences, such as 40 or 50 tasks. Experiments with larger task numbers on different datasets are essential to better demonstrate scalability and generalization ability.
5.	Some recent and competitive CIL methods are missing from the comparison, including TUNA [1], MOAL [2], and MIN [3].

References
[1] Integrating Task-Specific and Universal Adapters for Pre-Trained Model-Based Class-Incremental Learning. ICCV 2025.
[2] Knowledge Memorization and Rumination for Pre-Trained Model-Based Class-Incremental Learning. CVPR 2025.
[3] Mixture of Noise for Pre-Trained Model-Based Class-Incremental Learning. NeurIPS 2025.

---

> ### Author Rebuttal · Authors · 2026-03-30
>
> Thank you for your careful review and valuable comments. We will revise the manuscript accordingly.
>
> **Empirical Evidence for Gradient Recycling**: Under the 20-task ImageNet-R setting, gradient norms show similar trends across tasks, so we use task 10 as a representative case. As shown in Table 1, the orthogonally projected update is much smaller than that of standard LoRA adaptation, while the recycled gradient remains close to the full gradient. This indicates that strict projection removes useful optimization signals, whereas gradient recycling preserves plasticity and improves adaptation.
>
> **Table 1.** Gradient norms during training on task 10 under the 20-task ImageNet-R setting.
> Method|1 ep|10 ep|30 ep|50 ep
> -|-|-|-|-
> **LoRA Adaptation**|0.024609|0.013137|0.005774|0.000431
> **Orthogonal**|0.001177|0.000545|0.000311|0.000023
> **GR-LoRA-Orthogonal**|0.000925|0.000428|0.000214|0.000017
> **GR-LoRA-Recycle**|0.024145|0.011166|0.005590|0.000451
>
> **Key Component Memory and Inference Latency**: Gradient Recycling is used only in backpropagation, introducing negligible overhead. The residual subspace is computed once per new task, following prior orthogonality-based methods[1,2]. LMS adds one extra forward pass through a sampled historical module during training, while GMS introduces moderate pseudo-sample generation cost during classifier alignment; both remain comparable to standard methods with proper batch-size control. Table 2 reports inference latency. Entropy-based selection mechanism has $O(T)$ inference cost, since the input needs to be evaluated by multiple task branches. This overhead occurs only at inference time, improves task discrimination, and is common in many CIL methods using module/expert selection[3]. In future work, we will further explore how to reduce this inference overhead. Overall, we believe that the trade-off between the additional inference cost and the performance gain is reasonable and does not affect the main conclusions of the paper.
>
> **Table 2.** Inference latency on the 50-task ImageNet-R benchmark.
> Method|Infer time per image(s)|Test Infer time (s)|$A_{Last}$
> -|-|-|-
> **MOS**|2.56|400.85|66.95
> **MACIL**|2.01|19.04|71.13
> **LoRA-DRS**|1.78|15.93|72.37
> **Tuna[3]**|2.71|402.04|75.23
> **GR-LoRA**|2.58|427.32|77.43
>
> **Stability under High Semantic Overlap**: In GR-LoRA, recycled non-orthogonal gradients are absorbed by task-specific LoRA modules rather than written into the shared parameter space. Thus, even if they contain information from previous tasks, their effect is confined to task-specific adaptations, reducing interference with shared representations. To examine the relatively smaller gain on CUB-200, we compare GR-LoRA with joint training under the same trainable-parameter budget. As shown in Table 3, GR-LoRA remains close to joint training, suggesting that the smaller gain mainly comes from the fine-grained nature of the dataset rather than reduced robustness.
>
> **Table 3.** Comparison between GR-LoRA and joint training accuracy on CUB-200 under the same trainable-parameter budget.
> Method|1-task|10-task|20-task|50-task
> -|-|-|-|-
> **Joint Training**|90.29|89.95|89.99|88.89
> **GR-LoRA**|-|89.91|89.76|89.68
>
> **Longer Task Sequences**: GR-LoRA has been evaluated on longer task sequences, including 50-task and 20-task setting. Table 1 in the main paper reports results on CIFAR-100 and ImageNet-R under both settings, and Appendix B.1, Table 5 reports results on CUB-200 and ImageNet-A, showing consistent gains across datasets. In addition, Table 4 below shows that GR-LoRA remains the top-performing method on the 40-task ImageNet-R benchmark.
>
> **Table 4.** Comparison with SOTA methods on the 40-task ImageNet-R benchmark.
> Method|$A_{Last}$|$A_{Avg}$
> -|-|-
> **LoRA-DRS**|73.53|79.98
> **MACIL**|73.23|80.25
> **GR-LoRA**|77.82|84.04
>
> **Comparison with Recent Methods**: We further compare GR-LoRA with recent competitive CIL methods, including TUNA [3], MoAL [4], and MiN [5], on the 20-task and 50-task ImageNet-R benchmarks. Table 5 shows that GR-LoRA achieves the best results in both settings.
>
> **Table 5.** Comparison with recent methods on the 20-task and 50-task ImageNet-R benchmarks.
> Method|20-task||50-task||
> -|-|-|-|-
> ||$A_{Last}$|$A_{Avg}$|$A_{Last}$|$A_{Avg}$
> **TUNA**|77.28|82.47|75.23|80.71
> **MOAL**|74.20|82.15|52.90|68.99
> **MIN**|78.68|83.83|76.13|82.25
> **GR-LoRA**|80.42|85.42|77.43|83.46
>
> **References**:\
> [1] Liu, et al. LoRA Subtraction for Drift-Resistant Space in Exemplar-Free Continual Learning. *CVPR*, 2025.\
> [2] Wu, et al. Navigating semantic drift in task-agnostic class-incremental learning. *ICML* 2025.\
> [3] Wang, et al. Integrating Task-Specific and Universal Adapters for Pre-Trained Model-Based Class-Incremental Learning. *ICCV*, 2025.\
> [4] Gao, et al. Knowledge Memorization and Rumination for Pre-Trained Model-Based Class-Incremental Learning. *CVPR*, 2025.\
> [5] Jiang, et al. Mixture of Noise for Pre-Trained Model-Based Class-Incremental Learning. *NeurIPS*, 2025.

---

> > ### Author Rebuttal · Reviewer_jt37 · 2026-04-01
> >
> > Thank you for your detailed responses and the newly updated results. Most of my concerns have been addressed. However, I noticed that the proposed method has relatively poor inference efficiency, even worse than models such as TUNA and MOS, which require network-level iterative inference. This raises a serious concern about whether the proposed method can be applied in real-world scenarios, where the number of tasks may exceed 100 or even 1000. In such cases, this inference speed would be unacceptable.
> >
> > In fact, this is a significant issue. Nevertheless, given the interesting design and the mostly satisfactory experimental results, I will keep the borderline acceptance score. I strongly recommend that the authors carefully consider efficiency when designing future class-incremental learning algorithms, as the current version seems to suffer from a serious efficiency drawback.

---

> > > ### Author Response · Authors · 2026-04-06
> > >
> > > Thank you very much for reading our rebuttal and for your recognition of our method. We sincerely appreciate your valuable suggestions and will carefully consider efficiency when designing future class-incremental learning algorithms, which perfectly aligns with our future research directions.

---

### Official Review · Reviewer_6TPS · 2026-03-10

**Soundness:** 3
**Presentation:** 3
**Significance:** 4
**Originality:** 4
**Overall Recommendation:** 5
**Confidence:** 4

**Summary:**

This paper proposes the GR-LoRA method, which balances the plasticity and stability of the model through a gradient cyclic mechanism. It utilizes entropy to select the optimal task-specific LoRA module and proposes two suppression strategies to reduce the activation of mismatched modules to ensure the reliability of the selection.

**Compliance With Llm Reviewing Policy:**

Affirmed.

**Key Questions For Authors:**

1. Regarding Formula 4, it is suggested that you further elaborate on how the feature space is mapped to the parameter gradient space.
2. Regarding Formula 6, it is recommended that you clearly specify what constitutes the "mild conditions"; this would assist readers in better understanding the details of the formula.
3. It is recommended that the Results Analysis section utilize comparative data analysis to compellingly demonstrate the effectiveness of the optimization efforts.
4. In the computational overhead analysis section, it is recommended to further refine the experimental design—specifically by incorporating metrics such as actual runtime into Table 4—to provide a more comprehensive reflection of the method's performance.

**Limitations:**

No. It is suggested that the authors further explain what unresolved problems and shortcomings still exist in the current method.

**Strengths And Weaknesses:**

Dear  Author,
Thank you for proposing the GR-LoRA method, which balances model plasticity and stability through a gradient-cycling mechanism.
Your paper is clearly structured; technically, it demonstrates sound logic, and the proposed method is practical. However, there remain several areas in the paper that could be improved. Regarding Equation 4, we suggest further elaborating on how the feature space is mapped to the parameter gradient space. Regarding Equation 6, we recommend explicitly clarifying what constitutes the "mild conditions"; this would help readers better understand the technical details of the equation.
In the experimental section, we suggest enhancing the results analysis by incorporating a comparative analysis of the data to compellingly demonstrate the effectiveness of the proposed optimization. Regarding the analysis of computational overhead, we recommend refining the experimental design—specifically, by supplementing Table 4 with metrics such as actual runtime to provide a more comprehensive reflection of the method's performance. Furthermore, we suggest offering a forward-looking perspective on the future development trends of this technology, thereby providing the paper with a broader academic scope.
Overall, your research work possesses significant academic value and broad prospects for future development. We sincerely recommend that you revise your paper in light of the aforementioned comments to further enhance its quality and readability. We look forward to receiving your revised manuscript and wish you even greater success in your future research endeavors.

---

> ### Author Rebuttal · Authors · 2026-03-30
>
> Thank you for your careful review and valuable comments. We will revise the manuscript accordingly.
>
> **Feature-to-Gradient Space Mapping in Formula (4)**:
> As established in existing work [1], for a standard linear layer, the parameter gradient space is fundamentally spanned by the input feature space. Specifically, for a linear layer $Z=XW$, the chain rule dictates that $\nabla_WL=X^T\nabla_ZL$. This equation inherently restricts the parameter gradients to be linear combinations of the input features $X$, meaning the feature space directly maps to and defines the gradient space. Building on this mathematical premise, applying the feature-derived orthogonal projector $P_t^l(P_t^l)^T$ directly to the gradient $g_{t,s}^l$ in our method is rigorous. It effectively constrains the parameter updates to the residual feature subspace of the current task, ensuring minimal interference with previously learned tasks, while the discarded non-orthogonal gradient components are recycled into task-specific modules to maintain plasticity.
>
> **Mild Conditions in Formula (6)**: The efficient formulation in Formula 6 follows the design of LoRA-FA [2]. Here, the mild conditions specifically mean that $A$ is randomly initialized from a normal distribution and kept fixed during training, while the output projection matrix $B$ is initialized to zero and optimized. Under this setting, the fixed random matrix $A$ forms a rank-$r$ subspace that serves as a stable basis. Optimizing only $B$ is therefore sufficient to approximate the updates obtained by jointly optimizing both $A$ and $B$, while significantly reducing computational cost.
>
> **Writing and Presentation**: We will revise the Results Analysis section to incorporate more explicit comparative data analysis, as suggested. This will provide clearer evidence of the effectiveness of the proposed optimization strategies across different experimental settings.
>
> **Inference Cost**: We further include actual runtime metrics, including inference time per image and total test inference time, as shown in Table 1, to complement experiments. The entropy-based selection mechanism has $O(T)$ inference cost, since the input needs to be evaluated by multiple task branches. This overhead occurs only at inference time, improves task discrimination, and is common in many mainstream CIL methods using module/expert selection[3]. In future work, we will further explore how to reduce this inference overhead. Overall, we believe that the trade-off between the additional inference cost and the performance gain is reasonable and does not affect the main conclusions of the paper.
>
> **Table 1.** Comparison on the 50-task ImageNet-R benchmark in terms of inference time cost.
> |Method|Infer time per image(s)|Test Infer time (s)|*$A_{Last}$*|
> |-|-|-|-|
> |**MOS**|2.56|400.85|66.95|
> |**MACIL**|2.01|19.04|71.13|
> |**LoRA-DRS**|1.78|15.93|72.37|
> |**Tuna[3]**|2.71|402.04|75.23|
> |**GR-LoRA (Ours)**|2.58|427.32|77.43|
>
> **References**: \
> [1] Wang, et al. Training networks in null space of feature covariance for continual learning. *CVPR*, 2021.\
> [2] Zhang, et al. LORA-FA: MEMORY-EFFICIENT LOW-RANK ADAPTATION FOR LARGE LANGUAGE MODELS FINE-TUNING. *ArXiv* 2023.\
> [3] Wang, et al. Integrating Task-Specific and Universal Adapters for Pre-Trained Model-Based Class-Incremental Learning. *ICCV*, 2025.

---

### Official Review · Reviewer_iLKq · 2026-03-11

**Soundness:** 3
**Presentation:** 2
**Significance:** 2
**Originality:** 3
**Overall Recommendation:** 4
**Confidence:** 4

**Summary:**

The manuscript tackles the stability-plasticity dilemma in Class-Incremental Learning. The core premise is that standard orthogonality-based CIL methods inherently choke model plasticity by discarding non-orthogonal gradient components. To bypass this bottleneck, the authors propose GR-LoRA, which explicitly recycles these discarded gradients into lightweight and task-specific modules. The routing problem during inference is handled via an entropy-based selection metric, supported by Local Mismatch Suppression during training and Global Mismatch Suppression through out-of-distribution feature synthesis post-training.

**Compliance With Llm Reviewing Policy:**

Affirmed.

**Final Justification:**

I have carefully evaluated the author rebuttal. The provided empirical data effectively addresses my initial concerns regarding the memory footprint and parameter capacity isolation. However, the inference latency remains a critical concern. Providing the second table was a necessary step for transparency, but a test inference time of 427 seconds for fifty tasks exposes a severe efficiency drawback. While I recognize the authors' argument that this linear computational scaling is a systemic limitation of current entropy-based routing architectures rather than a flaw unique to this specific design, it remains a major bottleneck for practical deployment. Given the strictly superior final accuracy achieved against parallel methods, I consider this severe computational trade-off to be marginally acceptable. Consequently, I am raising my overall recommendation score to 4. I recommend that the authors incorporate the inference latency results from the second table and an explicit discussion of this efficiency limitation into the main text of the final version to ensure transparency for future readers.

**Key Questions For Authors:**

-What is the exact memory footprint required to store the class prototypes and high-dimensional covariance matrices for all past classes across a long task sequence?

-How does the inference latency and computational overhead scale empirically when the number of tasks reaches 50 or more?

-Will you provide the missing baseline results comparing GR-LoRA to a single LoRA module with an equivalent total parameter capacity?

**Limitations:**

-Table 4 offers a starting point for discussing computational overhead at 20 tasks. The authors must explicitly confront the inference latency and linear MAC scaling for longer sequences like 50 tasks.

-The memory overhead induced by storing high-dimensional covariance matrices for GMS needs to be honestly acknowledged as a strict deployment constraint.

**Strengths And Weaknesses:**

**Strengths**

-Decomposing gradients via a residual subspace projection matrix is mathematically rigorous. Routing different gradient components to separate parameter sets using a dual-component LoRA is a neat engineering trick perfectly aligned with the main objective. Introducing LMS and GMS effectively plugs the module mis-selection hole prevalent in routing-based architectures.

-Theorems 3.1 and 3.2 do a good job formalizing the plasticity gap caused by strict orthogonal constraints.

-The empirical results in Table 2 are strong and show consistent gains over recent baselines across multiple benchmarks. The ablation studies in Table 3 and Figure 4 transparently isolate the specific contributions of the proposed mismatch suppression components.


**Weaknesses**
-Figure 2 is far too dense. It tries to cram in too many mechanisms at once, making the visual hierarchy between Gradient Recycling, LMS, and GMS incredibly hard to parse.
-GMS relies on storing the class prototype $\mu_c$ and covariance matrix for all past classes to synthesize OOD features. The authors do not adequately address the memory footprint of storing these high-dimensional covariance matrices compared to strict exemplar-free CIL methods.
-The manuscript heavily relies on the entropy-based metric formulated in Equation 8 for module selection during inference. This mechanism requires passing the input through all $T$ historical modules to calculate the prediction entropy. Consequently, as the number of tasks $T$ grows to larger scales such as 50 tasks, the inference computational cost scales linearly with $T$. While Table 4 demonstrates MACs for a 20-task setting, it fails to explicitly analyze the inference latency and MAC scaling for sequences of 50 tasks or more.
-The ablation study presented in Table 3 demonstrates performance improvements from the recycling mechanism but fundamentally fails to isolate these gains from the mere increase in trainable parameters. A critical missing baseline is a single adaptation module possessing an equivalent total parameter count.

---

> ### Author Rebuttal · Authors · 2026-03-30
>
> Thank you for your careful review and valuable comments. We will revise the manuscript accordingly.
>
> **Writing and Presentation**: We will revise Figure 2 to simplify its layout and improve clarity. In addition, we will provide a clearer and more detailed explanation of LMS and GMS. We will also thoroughly proofread the manuscript to correct spelling and typographical errors.
>
> **Storage Overhead**: Storing class prototypes and covariance statistics is a common and effective strategy in CIL, as it helps mitigate the bias of the classifier toward newly learned tasks and improves the retention of previously acquired knowledge [1,2]. Compared with standard prototype-based classifier[3] or classifier alignment[4,5] methods, GR-LoRA introduces only negligible additional storage overhead. Specifically, our method requires storing one extra shared-space prototype $\mu_c \in \mathbb{R}^{1 \times 768}$ per class. This additional prototype is used only for constructing OOD prototypes, as described in Eq. (10) and Eq. (11). Importantly, GR-LoRA does not require storing any extra covariance matrix. Following MACIL [6], the generated OOD samples reuse the same private-space covariance matrix $\hat{\Sigma}_c$ as the original ID samples. As shown in Table 1, the storage cost of GR-LoRA is 2.365 MB per class, compared with 2.362 MB for standard Classify Alignment. Therefore, the additional overhead is only about 3 KB per class, which is negligible in practice.
>
> **Table 1.** Storage overhead comparison of representative prototype-based CIL methods per class with feature dimension $d = 768$.
> |Method|Stored Variables|Parameters|Storage(MB)|
> |-|-|-|-|
> |**Prototype Classifier**|$\mu$|$d$|0.003|
> |**Classifier Alignment**|$\mu$+$\Sigma$|$d+d^2$|2.362|
> |**GR-LoRA (Ours)**|$\mu+\hat{\mu}+\hat{\Sigma}$|$2d+d^2$|2.365|
>
> **Inference and Computation Cost**: The MACs reported in Table 4 of the original paper measure the total computation of a single forward pass, including the ViT backbone, the shared LoRA, and the task-specific LoRA, and are intended to reflect the computational efficiency of the model. We further report the 50-task MACs and inference latency results in Table 2. We acknowledge that the entropy-based selection mechanism has $O(T)$ inference cost, since the input needs to be evaluated by multiple task branches. This overhead occurs only at inference time, improves task discrimination, and is common in many mainstream CIL methods using module/expert selection[5]. In future work, we will further explore how to reduce this inference overhead. Overall, we believe that the trade-off between the additional inference cost and the performance gain is reasonable and does not affect the main conclusions of the paper.
>
> **Table 2.** Comparison on the 50-task ImageNet-R benchmark in terms of MACs and inference time cost.
> |Method|MACs(G)|Infer time per image(s)|Test Infer time (s)|*$A_{Last}$*|
> |-|-|-|-|-|
> |**MOS**|16.92|2.56|400.85|66.95|
> |**MACIL**|28.48|2.01|19.04|71.13|
> |**LoRA-DRS**|20.50|1.78|15.93|72.37|
> |**Tuna[5]**|16.92|2.71|402.04|75.23|
> |**GR-LoRA (Ours)**|20.56|2.58|427.32|77.43|
>
> **Effect of Parameter Scaling**: To isolate the effect of the proposed recycling mechanism from parameter scaling, we already include parameter-matched and ablation comparisons. As shown in Table 3 of the original paper, Single LoRA assigns independent LoRA modules to each task without any constraints while using the same entropy-based selection strategy, yet performs substantially worse than GR-LoRA. Moreover, when $r = 5$, GR-LoRA matches the parameter count of the standard single-branch LoRA baseline at 0.38M and still achieves superior performance, as shown in Table 4 of the original paper. These results indicate that the performance gains primarily arise from the proposed gradient-recycling mechanism rather than increased model capacity.
>
> **References**:\
> [1] Chen, et al. Adaptive Retention \& Correction: Test-Time Training for Continual Learning. *ICLR*, 2025.\
> [2] Grzegorz, et al. Task-recency bias strikes back: Adapting covariances in Exemplar-Free Class Incremental Learning. *NeurIPS*, 2024.\
> [3] Liu, et al. LoRA Subtraction for Drift-Resistant Space in Exemplar-Free Continual Learning. *CVPR*, 2025.\
> [4] Sun, et al. Mos: Model surgery for pre-trained modelbased class-incremental learning. *AAAI*, 2025.\
> [5] Wang, et al. Integrating Task-Specific and Universal Adapters for Pre-Trained Model-based Class-Incremental Learning. *ICCV*, 2025.\
> [6] Wu, et al. Navigating semantic drift in task-agnostic class-incremental learning. *ICML* 2025.

---

> > ### Author Rebuttal · Reviewer_iLKq · 2026-04-03
> >
> > I have carefully evaluated the author rebuttal.
> > The provided empirical data effectively addresses my initial concerns regarding the memory footprint and parameter capacity isolation.
> > However, the inference latency remains a critical concern. Providing the second table was a necessary step for transparency, but a test inference time of 427 seconds for fifty tasks exposes a severe efficiency drawback.
> > While I recognize the authors' argument that this linear computational scaling is a systemic limitation of current entropy-based routing architectures rather than a flaw unique to this specific design, it remains a major bottleneck for practical deployment. Given the strictly superior final accuracy achieved against parallel methods, I consider this severe computational trade-off to be marginally acceptable.
> > Consequently, I am raising my overall recommendation score to 4.
> > I recommend that the authors incorporate the inference latency results from the second table and an explicit discussion of this efficiency limitation into the main text of the final version to ensure transparency for future readers.

---

> > > ### Author Response · Authors · 2026-04-06
> > >
> > > Thank you very much for reading our rebuttal and for your recognition of our method. We sincerely appreciate your valuable suggestions and will incorporate the inference latency results and an explicit discussion of this efficiency limitation into the final version of our manuscript to ensure transparency for future readers.

---

### Official Review · Reviewer_mYKe · 2026-03-13

**Soundness:** 3
**Presentation:** 3
**Significance:** 3
**Originality:** 4
**Overall Recommendation:** 5
**Confidence:** 4

**Summary:**

The paper proposes Gradient-Recycling Low-Rank Adaptation (GR-LoRA) for Class-Incremental Learning. The authors argue that existing orthogonality-based methods project gradients onto subspaces that are orthogonal to past tasks to prevent interference, but this explicitly discards non-orthogonal components and progressively reduces the optimization space, hence hindering plasticity over long task sequences. GR-LoRA aims to solve this by recycling these discarded non-orthogonal gradient components into task-specific lightweight modules.

During inference, the method uses prediction entropy to select the optimal task-specific module. To prevent old modules from over-generalizing and producing overconfident predictions on current data, the authors incorporate two suppression strategies. Local Mismatch Suppression (LMS) treats current data as OOD relative to old modules during the training phase, whereas Global Mismatch Suppression (GMS) generates synthetic OOD features from old prototypes to re-calibrate global decision boundaries during classifier alignment. The authors provide a theoretical analysis and test GR-LoRA on standard CIL benchmarks.

**Compliance With Llm Reviewing Policy:**

Affirmed.

**Final Justification:**

The authors resolved all my concerns in the rebuttal.

**Key Questions For Authors:**

Some typos:
 - "Module Setect Acc" instead of "Select" in Figure 4,
 -  "OOD dection" instead of "detection" in Section 4.3
 -  Extra comma after the LoRaformula at line 173

**Limitations:**

yes

**Strengths And Weaknesses:**

## Strengths
1. The idea of differentiating between orthogonal and non-orthogonal subspace projectors, and explicitly recycling the discarded non-orthogonal components, is novel and interesting.
2. The paper is well-written, logically structured, and generally easy to follow.
3. The method provides SOTA performance across multiple established Class-Incremental Learning benchmarks, proving its practical effectiveness.

## Weaknesses
1. Figure 2 is quite messy and visually overwhelming. It attempts to convey too much information and would greatly benefit from being simplified or divided into separate figures. The explanations for LMS and GMS are somewhat brief and could be expanded for clarity. Conversely, the theoretical analysis in the main text feels not fully necessary and could be moved to the supplementary material to free up valuable space.
2. The ablation studies are currently limited only to the ImageNet-R dataset. Evaluating the individual components across more datasets is necessary to fully prove their generalizability.
3. The memory and computational complexity analysis is not fully satisfactory. Since the method allocates task-specific modules, the memory complexity scales at $O(N)$ with the number of tasks. Even if the modules are lightweight, a theoretical and experimental comparison regarding memory overhead over long task sequences is necessary to improve fairness.
4. The experimental evaluation is missing some recent baselines (e.g., RanPAC, McDonnell 2023), which should be included.

**If the main points are addressed during rebuttal, I will raise my score to accept.**

---

> ### Author Rebuttal · Authors · 2026-03-30
>
> Thank you for your careful review and valuable comments. We will revise the manuscript accordingly.
>
> **Writing and Presentation**: We will revise Figure 2 to simplify its layout and improve clarity. In addition, we will reorganize the presentation of the theoretical analysis in the main text and provide a clearer and more detailed explanation of LMS and GMS. We will also thoroughly proofread the manuscript to correct spelling and typographical errors.
>
> **Ablation Study**: Additional ablation studies are conducted on CIFAR-100 and ImageNet-A under the 20-task CIL setting. As shown in Table 1, progressively incorporating the proposed components consistently improves performance across both datasets, validating the effectiveness of each module and demonstrating the generalizability of GR-LoRA under diverse and more challenging data distributions.
>
> **Table 1** Ablation study of individual component contributions on CIFAR-100 and ImageNet-A under the 20-task CIL setting.
> |**Method**|**CA**|**LMS**|**GMS**|**CIFAR-100**||**ImageNet-A**||
> |-|-|-|-|-|-|-|-|
> |||||$A_{Last}$|$A_{Avg}$|$A_{Last}$|$A_{Avg}$|
> |**LoRA-DRS**|✓|✗|✗|90.34|93.93|55.69|64.39|
> |**Single LoRA**|✓|✓|✓|88.06|92.41|58.53|64.38|
> |**GR-LoRA (Ours)**|✓|✗|✗|90.41|94.26|58.20|66.71|
> ||✓|✓|✗|90.74|94.45|59.12|67.91|
> ||✓|✗|✓|91.25|94.77|61.29|70.09|
> ||✓|✓|✓|**91.42**|**94.84**|**62.08**|**70.55**|
>
> **Memory and Computational Complexity**: Memory complexity scales as $O(N)$. To ensure a fair comparison over long task sequences, both additional memory cost (excluding the ViT backbone) and MACs under the 50-task setting are reported in Table 2, including per-task and total memory. Additional storage overhead arises from task-specific modules and grows linearly with the number of tasks. This overhead can be mitigated by reducing the LoRA rank, and the method is not sensitive to the rank choice, as shown in Table 4 of the original paper.
>
> **Table 2.** Comparison on the 50-task ImageNet-R benchmark in terms of MACs and Memory cost.
> |Method|MACs(G)|Memory cost per task(MB)|Total memory cost(MB)|*$A_{Last}$*|
> |-|-|-|-|-|
> |**MOS**|16.92|1.23|61.56|66.95|
> |**MACIL**|28.48|4.50|225.00|71.13|
> |**LoRA-DRS**|20.50|1.41|70.31|72.37|
> |**Tuna[1]**|16.92|2.46|62.79|75.23|
> |**GR-LoRA (Ours)**|20.56|2.81|140.62|77.43|
>
> **Comparison with Recent Baseline**: We include additional comparisons with RanPAC [2] under the same experimental settings on the 20-task and 50-task ImageNet-R benchmarks. As shown in Table 3, our method consistently outperforms RanPAC across both task settings, with more pronounced gains in the longer task setting. These results further substantiate the effectiveness of GR-LoRA.
>
> **Table 3.** Comparison with RanPAC [2] on the 20-task and 50-task ImageNet-R benchmarks.
> |Method||ImageNet-R|||
> |-|-|-|-|-|
> ||20-task||50-task||
> ||*$A_{Last}$*|*$A_{Avg}$*|*$A_{Last}$*|*$A_{Avg}$*|
> |**RanPAC**|77.45|82.71|73.67|79.75|
> |**GR-LoRA (Ours)**|**80.42**|**85.42**|**77.43**|**83.46**|
>
> **References**: \
> [1] Wang, et al. Integrating Task-Specific and Universal Adapters for Pre-Trained Model-Based Class-Incremental Learning. *ICCV*, 2025.\
> [2] McDonnell, et al. RanPAC: Random Projections and Pre-trained Models for Continual Learning. *NeurIPS*, 2023.

---

> > ### Author Rebuttal · Reviewer_mYKe · 2026-04-04
> >
> > Thank you for your rebuttal. My concerns are fully resolved.

---

> > > ### Author Response · Authors · 2026-04-06
> > >
> > > Thank you very much for reading our rebuttal and for your recognition of our method. We are pleased to hear that your concerns are fully resolved. We sincerely appreciate your valuable suggestions and will revise the manuscript accordingly.

---

### Decision · Program_Chairs · 2026-04-30

**Decision:**

Accept (spotlight)

**Comment:**

The paper received original score of (3,4,5,4). The authors provided rebuttal which convinced two reviewers to raise the score (4,5,5,4) resulting in consensus for acceptance. There was a discussion amongst reviewers on efficiency of the method (inference latency) which was considered the main drawback of the method. However, reviewers considers that the strengths of the methods clearly outweighed this drawback. The AC agrees and recommends acceptance.

The authors should carefully prepare the final version and include the additional results provided in the rebuttal in the final version.